# LogicMP: A Neuro-symbolic Approach for Encoding First-order Logic Constraints

**Weidi Xu**[1,2], **Jingwei Wang**[2], **Lele Xie**[2], **Jianshan He**[2], **Hongting Zhou**[2],
**Taifeng Wang**[3], **Xiaopei Wan**[2], **Jingdong Chen**[2], **Chao Qu**[1], **Wei Chu**[2,1] *
[1]INFLY TECH (Shanghai) Co., Ltd [2]Ant Group [3]BioMap Research
`wdxu@inftech.ai`

## Abstract

Integrating first-order logic constraints (FOLCs) with neural networks is a crucial but challenging problem since it involves modeling intricate correlations to satisfy the constraints. This paper proposes a novel neural layer, LogicMP, which performs mean-field variational inference over a Markov Logic Network (MLN). It can be plugged into any off-the-shelf neural network to encode FOLCs while retaining modularity and efficiency. By exploiting the structure and symmetries in MLNs, we theoretically demonstrate that our well-designed, efficient mean-field iterations greatly mitigate the difficulty of MLN inference, reducing the inference from sequential calculation to a series of parallel tensor operations. Empirical results in three kinds of tasks over images, graphs, and text show that LogicMP outperforms advanced competitors in both performance and efficiency.

## 1 Introduction

The deep learning field has made remarkable progress in the last decade, owing to the creation of neural networks (NNs) (Goodfellow et al., 2016; Vaswani et al., 2017). They typically use a feed-forward architecture, where interactions occur implicitly in the middle layers with the help of various neural mechanisms. However, these interactions do not explicitly impose logical constraints among prediction variables, resulting in predictions that often do not meet the structural requirements.

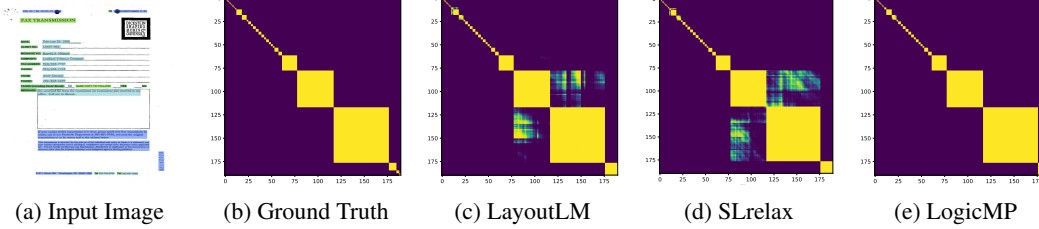

|  (a) Input Image | (b) Ground Truth | (c) LayoutLM | (d) SLrelax | (e) LogicMP |

Figure 1: The document understanding task predicts whether every two tokens coexist in the same block in an input document image (**a**). The FOLC regarding the transitivity of coexistence can be used to obtain the structured output. The ground truth (**b**) typically forms several squares for the segments. Both NN (Xu et al., 2020) (**c**) and advanced method (Xu et al., 2018) (**d**) struggle to meet the FOLC where many `coexist` variables are incorrectly predicted. In contrast, LogicMP (**e**) is effective while maintaining modularity and efficiency. See Sec. 5 for complete experimental details.

This paper investigates the problem of incorporating *first-order logic constraints* (FOLCs) into neural networks. An example of FOLCs can be found in the document understanding task (Jaume et al., 2019), which aims to segment the given tokens into blocks for a document image (Fig. 1(a)). We formalize the task into the binary coexistence prediction of token pairs $C(a, b) \in \{0, 1\}$ where $C$ denotes the co-existence of tokens $a, b$ (Fig. 1(b)). There is a FOLC about the transitivity of coexistence predictions: when tokens $a$ and $b$ coexist in the same block, and tokens $b$ and $c$ coexist in

---

*Corresponding author. Code is available at: `https://github.com/wead-hsu/logicmp`

Figure 2: **A high-level view of LogicMP.** NNs typically use the softmax layer for independent prediction (**left**), which can be replaced by a LogicMP encoding FOLCs (**middle**). LogicMP is implemented (**right**) by efficient mean-field iterations which leverage the structure of MLN (Sec. 3).

the same block, then $a$ and $c$ must coexist, i.e., $\forall a, b, c : \mathtt{C}(a, b) \wedge \mathtt{C}(b, c) \implies \mathtt{C}(a, c)$, to which we refer as "transitivity rule". NNs generally predict $\mathtt{C}(\cdot, \cdot)$ independently, failing to meet this FOLC (Fig. 1(c)), and the same applies to the advanced regularization method (Xu et al., 2018) (Fig. 1(d)). We aim to incorporate the transitivity rule into NNs so that the predicted result satisfies the logical constraint (Fig. 1(e)). FOLCs are also critical in many other real-world tasks, ranging from collective classification tasks over graphs (Richardson & Domingos, 2006; Singla & Domingos, 2005) to structured prediction over text (Sang & Meulder, 2003).

Incorporating such FOLCs into neural networks is a long-standing challenge. The main difficulty lies in modeling intricate variable dependencies among massive propositional groundings. For instance, for the transitivity rule with 512 tokens, 262K coexistence variables are mutually dependent in 134M groundings. Essentially, modeling FOLCs involves the weighted first-order model counting (WFOMC) problem, which has been extensively studied in the previous literature (den Broeck & Davis, 2012; Dalvi & Suciu, 2013; Gribkoff et al., 2014). However, it is proved #P-complete for even moderately complicated FOLCs (Dalvi & Suciu, 2013), such as the transitivity rule mentioned above.

Markov Logic Networks (MLNs) (Richardson & Domingos, 2006) are a common approach to modeling FOLCs, which use joint potentials to measure the satisfaction of the first-order logic rules. MLN is inherited from WFOMC, and is difficult to achieve exact inference (Gribkoff et al., 2014). Although MLN formalization allows for approximate inference, MLNs have long suffered from the absence of efficient inference algorithms. Existing methods typically treat the groundings individually and fail to utilize the structure and symmetries of MLNs to accelerate computation (Yedidia et al., 2000; Richardson & Domingos, 2006; Poon & Domingos, 2006). ExpressGNN (Qu & Tang, 2019; Zhang et al., 2020) attempts to combine MLNs and NNs using variational EM, but they remain inherently independent due to the inference method's inefficiency. Some lifted algorithms exploit the structure of MLNs to improve efficiency but are infeasible for neural integration due to their requirements of symmetric input (de Salvo Braz et al., 2005; Singla & Domingos, 2008; Niepert, 2012) or specific rules (den Broeck & Davis, 2012; Gribkoff et al., 2014).

This paper proposes a novel approach called ***Logical Message Passing*** (LogicMP) for general-purpose FOLCs. It is an efficient MLN inference algorithm and can be seamlessly integrated with any off-the-shelf NNs, positioning it as a neuro-symbolic approach. Notably, it capitalizes on the benefits of parallel tensor computation for efficiency and the plug-and-play principle for modularity. Fig. 2 illustrates the computational graph of LogicMP. Bare NNs (Fig. 2a) predict each output variable independently. LogicMP can be stacked on top of any encoding network as an efficient modular neural layer that enforces FOLCs in prediction (Fig. 2b). Specifically, LogicMP introduces an efficient mean-field (MF) iteration algorithm for MLN inference (Fig. 2c). This MF algorithm enables LogicMP's outputs to approximate the variational approximation of MLNs, ensuring that FOLCs can be encoded into LogicMP's inputs. In contrast to vanilla MF algorithms that rely on inefficient sequential operations (Wainwright & Jordan, 2008; Koller & Friedman, 2009), our well-designed MF iterations can be formalized as Einstein summation, thereby supporting parallel tensor computation. This formalization benefits from our exploitation of the structure and symmetries of MLNs (Sec. 3.2), which is supported by theoretical guarantees (Sec. 3.1).

In Sec. 5, we demonstrate the versatility of LogicMP by evaluating its performance on various real-world tasks from three domains: visual document understanding over images, collective classification over graphs, and sequential labeling over text. First, we evaluate LogicMP on a real-world document understanding benchmark dataset (FUNSD) (Jaume et al., 2019) with up to 262K mutually-dependent variables and show that it outperforms previous state-of-the-art methods (Sec. 5.1). Notably, the results demonstrate that LogicMP can lead to evident improvements even when imposing a single

FOLC on prediction variables, which is beyond the capacity of existing methods using arithmetic circuits (ACs). For the second task (Sec. 5.2), we conduct experiments on relatively large datasets in the MLN literature, including UW-CSE (Richardson & Domingos, 2006) and Cora (Singla & Domingos, 2005). Our results show that LogicMP significantly speeds up by about 10x compared to competitive MLN inference methods, which enables larger-scale training for better performance. Finally, we evaluate LogicMP on a sequence labeling task (CoNLL-2003) (Sang & Meulder, 2003) and show that it can leverage task-specific rules to improve performance over competitors (Sec. 5.3).

**Contributions.** We: (*i*) Present a novel, modular, and efficient neural layer LogicMP, the first fully differentiable neuro-symbolic approach capable of encoding FOLCs for arbitrary neural networks. (*ii*) Design an accelerated mean-field algorithm for MLN inference that leverages the structure and symmetries in MLNs, formalizing it to parallel computation with a reduced complexity from $\mathcal{O}(N^M L^2 D^{L-1})$ to $\mathcal{O}(N^{M'} L^2)$ $(M' \leq M)$ (Sec. 3). For instance, LogicMP can incorporate FOLCs with up to 262K variables within 0.03 seconds, where AC-based methods fail during compilation. (*iii*) Demonstrate its effectiveness and versatility in challenging tasks over images, graphs, and text, where LogicMP outperforms state-of-the-art neuro-symbolic approaches, often by a noticeable margin.

## 2 MARKOV LOGIC NETWORKS

An MLN is built upon a knowledge base (KB) $\{E, R, O\}$ consisting of a set $E = \{e_k\}_k$ of entities, a set $R = \{r_k\}_k$ of predicates, and a set $O$ of observation. Entities are also called constants (e.g., tokens). Each **predicate** represents a property or a relation, e.g., coexist (C). With particular entities assigned to a predicate, we obtain a **ground atom**, e.g., $C(e_1, e_2)$ where $e_1$ and $e_2$ are two tokens. For a ground atom $i$, we use a random variable $v_i$ in the MLN to denote its status, e.g., $v_{C(e_1,e_2)} \in \{0, 1\}$ denoting whether $e_1$ and $e_2$ coexist. The MLN is defined over all variables $\{v_i\}_i$ and a set of first-order logic formulas $F$. Each formula $f \in F$ represents the correlation among the variables, e.g., $\forall a, b, c : C(a, b) \wedge C(b, c) \implies C(a, c)$ which equals to $\forall a, b, c : \neg C(a, b) \vee \neg C(b, c) \vee C(a, c)$ by De Morgan's law. With particular entities assigned to the formula, we obtain a ground formula, aka **grounding**, e.g., $\neg C(e_1, e_2) \vee \neg C(e_2, e_3) \vee C(e_1, e_3)$. For a grounding $g$, we use $\mathbf{v}_g$ to denote the variables in $g$, e.g., $\{v_{C(e_1,e_2)}, v_{C(e_1,e_2)}, v_{C(e_1,e_3)}\}$. In MLN, each $f$ is associated with a weight $w_f$ and a potential function $\phi_f(\cdot) : \mathbf{v}_g \mapsto \{0, 1\}$ that checks whether the formula is satisfied in $g$. For each formula $f$, we can obtain a set of groundings $G_f$ by enumerating all assignments. We adopt the open-world assumption (OWA) and jointly infer all **unobserved facts**.

Based on the KB and formulas, we express the MLN as follows:

$$p(\mathbf{v}|O) \propto \exp\Big( \underbrace{\sum_i \phi_u(v_i)}_{neural\ semantics} + \underbrace{\sum_{f \in F} w_f \sum_{g \in G_f} \phi_f(\mathbf{v}_g)}_{symbolic\ FOLCs} \Big), \tag{1}$$

where $\mathbf{v}$ is the set of unobserved variables. The second term is for symbolic FOLCs, where $\sum_{g \in G_f} \phi_f(\mathbf{v}_g)$ measures the number of satisfied groundings of $f$. We explicitly express the first term to model the evidence of single ground atom $i$ in status $v_i$ using the unary potential $\phi_u(\cdot) : v_i \mapsto \mathcal{R}$. By parameterizing $\phi_u$ with an NN, this formulation enables semantic representation, allowing external features, such as pixel values of an image, to be incorporated in addition to the KB. Note that $\phi_u$ varies with different $i$, but for the sake of simplicity, we omit $i$ in the notation.

### 2.1 MEAN-FIELD ITERATION FOR MLN INFERENCE

The MLN inference is a persistent and challenging problem, as emphasized in (Domingos & Lowd, 2019). In an effort to address this issue, we draw inspiration from CRFasRNN (Zheng et al., 2015) and employ the MF algorithm (Wainwright & Jordan, 2008; Koller & Friedman, 2009) to mitigate the inference difficulty, which breaks down the Markov network inference into multiple feed-forward iterations. Unlike the variational EM approach (Zhang et al., 2020), which requires additional parameters, MF does not introduce any extra parameters to the model.

We focus on the MLN inference problem with a fixed structure (i.e., rules). The MF algorithm is used for MLN inference by estimating the marginal distribution of each unobserved variable. It computes a variational distribution $Q(\mathbf{v})$ that best approaches $p(\mathbf{v}|O)$, where $Q(\mathbf{v}) = \prod_i Q_i(v_i)$ is a product

of independent marginal distributions over all unobserved variables. Specifically, it uses multiple **mean-field iterations** to update all $Q_i$ until convergence. Each mean-field iteration updates the $Q_i$ in closed-form to minimize $D_{KL}(Q(\mathbf{v})||p(\mathbf{v}|O))$ as follows (see derivation in App. A):

$$Q_i(v_i) \leftarrow \frac{1}{Z_i} \exp(\phi_u(v_i) + \sum_{f \in F} w_f \sum_{g \in G_f(i)} \hat{Q}_{i,g}(v_i)), \tag{2}$$

where $Z_i$ is the partition function, $G_f(i)$ is the groundings of $f$ that involve the ground atom $i$, and

$$\hat{Q}_{i,g}(v_i) \leftarrow \sum_{\mathbf{v}_{g-i}} \phi_f(v_i, \mathbf{v}_{g-i}) \prod_{j \in g_{-i}} Q_j(v_j) \tag{3}$$

is the **grounding message** that conveys information from the variables $g_{-i}$ to the variable $i$ w.r.t. the grounding $g$. $g_{-i}$ denotes the ground atoms in $g$ except $i$, e.g., $g_{-\texttt{C}(e_1,e_3)} = \{\texttt{C}(e_1, e_2), \texttt{C}(e_2, e_3)\}$.

## 2.2 Computational Complexity Analysis

**Complexity Notation.** Although MF simplifies MLN inference, vanilla iteration remains computationally expensive, with its exponential complexity in the arity and length of formulas. Let us examine the time complexity of a single iteration using Eq. 2. Denote $N$ as the number of constants in $E$, $M = \max_f |\mathcal{A}^f|$ as the maximum arity of formulas, $L = \max_f |f|$ as the maximum length (number of atoms) of formulas, and $D$ as the maximum number of labels of predicates (for typical binary predicates, $D = 2$; while for multi-class predicates in many tasks, $D$ may be large).

**Expectation calculation of grounding message.** The computation of grounding message $\hat{Q}_{i,g}(v_i)$ in Eq. 3 involves multiplying $\prod_{j \in g_{-i}} Q_j(v_j)$ (which is $\mathcal{O}(L)$) for all possible values of $\mathbf{v}_{g-i}$ (which is $\mathcal{O}(D^{L-1})$), resulting in a complexity of $\mathcal{O}(LD^{L-1})$. When $D$ is large, $D^{L-1}$ is essential.

**Aggregation of massive groundings.** Since the number of groundings $|G_f|$ is $\mathcal{O}(N^M)$, and a grounding generates grounding messages for all involved variables, we have $\mathcal{O}(N^M L)$ grounding messages. With the complexity of computing a grounding message being $\mathcal{O}(LD^{L-1})$, the total time complexity of an MF iteration in Eq. 2 is $\mathcal{O}(N^M L^2 D^{L-1})$, which is exponential in $M$ and $L$.

## 3 Efficient Mean-field Iteration via LogicMP

We make two non-trivial improvements on the vanilla MF iteration, enabling LogicMP to perform efficient MLN inference. (1) We find that the calculation of a single grounding message in Eq. 3 contains considerable unnecessary computations and its time complexity can be greatly reduced (Sec. 3.1). (2) We further exploit the structure and symmetries in MLN to show that the grounding message aggregation in Eq. 2 can be formalized with Einstein summation notation. As a result, MF iterations can be efficiently implemented via parallel tensor operations, which fundamentally accelerates vanilla sequential calculations (Sec. 3.2). In the following, we will introduce several concepts of mathematical logic, such as clauses and implications (see more details in App. B).

### 3.1 Less Computation per Grounding Message

**Clauses** are the basic formulas that can be expressed as the disjunction of literals, e.g., $f := \forall a, b, c : \neg\texttt{C}(a, b) \lor \neg\texttt{C}(b, c) \lor \texttt{C}(a, c)$. For convenience, we explicitly write the clause as $f(\cdot; \mathbf{n})$ where $n_i$ is the preceding negation of atom $i$ in the clause $f$, e.g., $n_{\texttt{C}(a,b)} = 1$ due to the $\neg$ ahead of $\texttt{C}(a, b)$. A clause corresponds to several equivalent **implications** where the premise implies the hypothesis, e.g., $\texttt{C}(a, b) \land \texttt{C}(b, c) \implies \texttt{C}(a, c)$, $\texttt{C}(a, b) \land \neg\texttt{C}(a, c) \implies \neg\texttt{C}(b, c)$, and $\texttt{C}(b, c) \land \neg\texttt{C}(a, c) \implies \neg\texttt{C}(a, b)$. Intuitively, the grounding message $\hat{Q}_{i,g}$ in Eq. 3 w.r.t. $g_{-i} \rightarrow i$ corresponds to an implication (e.g., $\texttt{C}(e_1, e_2) \land \texttt{C}(e_2, e_3) \implies \texttt{C}(e_1, e_3)$). Since the grounding affects $i$ only when the premise $g_{-i}$ is true, most assignments of $\mathbf{v}_{g-i}$ that result in false premises can be ruled out in $\sum_{\mathbf{v}_{g-i}}$ in Eq. 3.

**Theorem 3.1.** *(Message of clause considers true premise only.) For a clause formula $f(\cdot; \mathbf{n})$, the MF iteration of Eq. 2 is equivalent for $\hat{Q}_{i,g}(v_i) \leftarrow \mathbf{1}_{v_i = \neg n_i} \prod_{j \in g_{-i}} Q_j(v_j = n_j)$.*

The proof can be found in App. C. Table 1 briefly illustrates the idea of the proof: for assignments of $g_{-i}$ resulting in false premises, the potential can be ruled out since it makes no difference for

Table 1: For the grounding message of $g$ w.r.t $C(e_1, e_2) \wedge C(e_2, e_3) \Rightarrow C(e_1, e_3)$, only one assignment of $g_{-C(e_1,e_3)}$ makes difference to $C(e_1, e_3)$, i.e., useful.

| $\phi_f(v_g)$ | | $v_{C(e_1,e_3)}$ | | useful |
| --- | --- | --- | --- | --- |
| | | 0 | 1 | |
| $g_{-C(e_1,e_3)}$ | $(0,0)$ | $1 = 1$ | | ✗ |
| | $(0,1)$ | $1 = 1$ | | ✗ |
| | $(1,0)$ | $1 = 1$ | | ✗ |
| | $(1,1)$ | $0 \neq 1$ | | ✓ |

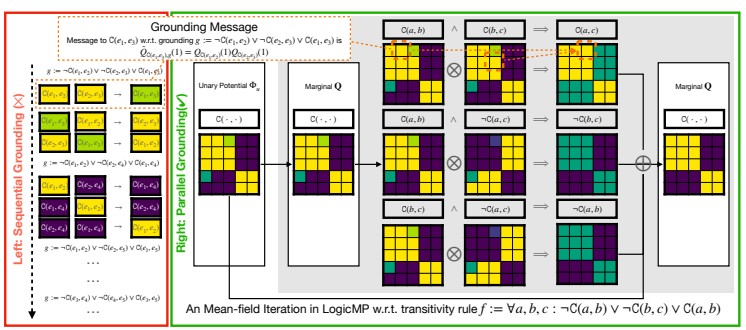

Figure 3: Instead of sequentially generating groundings (**left**), we exploit the structure of rules and formalize the MF iteration into Einstein summation notation, which enables parallel computation (**right**).

various assignments of the hypothesis $i$. Therefore, only the true premise $\{v_j = n_j\}_{j \in g_{-i}}$ needs to be considered. Compared to Eq. 3, the complexity is reduced from $\mathcal{O}(LD^{L-1})$ to $\mathcal{O}(L)$. The formulas in conjunctive normal form (CNF) are the conjunction of clauses. The simplification can also be generalized to CNF for $\mathcal{O}(L)$ complexity. The following theorem demonstrates this claim:

**Theorem 3.2.** *(Message of CNF = $\sum$ message of clause.) For a CNF formula with distinct clauses $f_k(\cdot; \mathbf{n})$, the MF iteration of Eq. 2 is equivalent for $\hat{Q}_{i,g}(v_i) \leftarrow \sum_{f_k} \mathbf{1}_{v_i = \neg n_i} \prod_{j \in g_{-i}} Q_j(v_j = n_j)$.*

See App. D for proof. This theorem indicates that the message of CNF can be decomposed into several messages of its clauses. Therefore, we only need to consider the clause formulas. We also generalize the theorem for the formulas with multi-class predicates to benefit general tasks (App. E).

## 3.2 PARALLEL AGGREGATION USING EINSTEIN SUMMATION

This subsection presents an efficient method for parallel message aggregation, i.e., $\sum_{g \in G_f(i)} \hat{Q}_{i,g}(v_i)$ in Eq. 2. In general, we can sequentially generate all propositional groundings of various formats in $G_f(i)$ to perform the aggregation. However, the number of possible groundings can be enormous, on the order of $\mathcal{O}(N^M)$, and explicitly generating all groundings is infeasible in space and time. By exploiting the structure of MLN and treating the grounding messages of the same first-order logic rule symmetrically, LogicMP automatically formalizes the message aggregation of first-order logic rules into *Einstein summation* (Einsum) notation. The Einsum notation indicates that aggregation can be achieved in parallel through tensor operations, resulting in acceleration at orders of magnitude.

The virtue lies in the summation of the product, i.e., $\sum_{g \in G_f(i)} \prod_{j \in g_{-i}} Q_j(v_j = n_j)$ by Theorem 3.1, which indicates that the grounding message corresponds to the implication from the premise $g_{-i}$ to the hypothesis $i$. Due to the structure of MLN, many grounding messages belong to the same implication and have the calculation symmetries, so we group them by their corresponding implications. The aggregation of grounding messages w.r.t. an implication amounts to integrating some rule arguments, and we can formalize the aggregation into Einsum. For instance, the aggregation w.r.t. the implication $\forall a, b, c : C(a, b) \wedge C(b, c) \implies C(a, c)$ can be expressed as $\texttt{einsum}(\text{"}ab, bc \to ac\text{"}, \mathbf{Q}_C(\mathbf{1}), \mathbf{Q}_C(\mathbf{1}))$ where $\mathbf{Q}_r(\mathbf{v}_r)$ denotes the collection of marginals w.r.t. predicate $r$, i.e., $\mathbf{Q}_r(\mathbf{v}_r) = \{Q_{r(\mathcal{A}_r)}(v_{r(\mathcal{A}_r)})\}_{\mathcal{A}_r}$ ($\mathcal{A}_r$ is the arguments of $r$). Fig. 3 illustrates this process, where we initially group the variables by predicates and then use them to perform aggregation using parallel tensor operations (see App. F) We formalize the parallel aggregation as follows:

**Proposition.** *Let $[f, h]$ denote the implication of clause $f$ with atom $h$ being the hypothesis and $\Phi_u(\mathbf{v}_r)$ denote the collection of $\phi_u(v_i)$ w.r.t. predicate $r$. For the grounding messages w.r.t. $[f, h]$ of a clause $f(\mathcal{A}^f; \mathbf{n}^f)$ to its atom $h$ with arguments $\mathcal{A}^f$, their aggregation is equivalent to:*

$$\check{\mathbf{Q}}_{r_h}^{[f,h]}(\mathbf{v}_{r_h}) \leftarrow \mathbf{1}_{\mathbf{v}_{r_h} = \neg n_h} \texttt{einsum}\left(\text{"}..., \mathcal{A}_{r_{j \neq h}}^f, ... \to \mathcal{A}_{r_h}^f\text{"}, ..., \mathbf{Q}_{r_{j \neq h}}(n_{j \neq h}), ...\right), \quad (4)$$

*where $r_h$ is the predicate of $h$, $\mathcal{A}_{r_h}^f$ is the arguments of $r_h$. The MF iteration of Eq. 2 is equivalent to:*

$$\mathbf{Q}_r(\mathbf{v}_r) \leftarrow \frac{1}{\mathbf{Z}_r} \exp\left(\Phi_u(\mathbf{v}_r) + \sum_{[f,h], r = r_h} w_f \check{\mathbf{Q}}_{r_h}^{[f,h]}(\mathbf{v}_{r_h})\right). \quad (5)$$

An additional benefit of using Einsum notation is it indicates a way to simplify complexity in practical scenarios. Let us consider a chain rule $\forall a, b, c, d \; : \; \mathtt{r_1}(a, b) \wedge \mathtt{r_2}(b, c) \wedge \mathtt{r_3}(c, d) \rightarrow \mathtt{r_4}(a, d)$. The complexity of $\mathtt{einsum}(\text{``}ab, bc, cd \rightarrow ad\text{''}, \mathbf{Q}_{\mathtt{r_1}}(\mathbf{1}), \mathbf{Q}_{\mathtt{r_2}}(\mathbf{1}), \mathbf{Q}_{\mathtt{r_3}}(\mathbf{1}))$ is $\mathcal{O}(N^4)$. By **Einsum optimization**, we can reduce it to $\mathcal{O}(N^3)$. Note that the Einsum optimization is almost free, as it can be done within milliseconds. This optimization method is not limited to chain rules and can be applied to other rules, which we demonstrate in App. G. For any rule, the optimized overall complexity is $\mathcal{O}(N^{M'}L^2)$ where $M'$ is the maximum number of arguments in the granular operations (App. H), in contrast to the original one $\mathcal{O}(N^M L^2 D^{L-1})$. In the worst case, $M'$ equals $M$, but in practice, $M'$ may be much smaller.

---

**Algorithm 1** LogicMP.

---

**Require:** Grouped unary potential $\{\Phi_u(\mathbf{v}_r)\}_r$, the formulas $\{f(\mathcal{A}; \mathbf{n})\}_f$ and rule weights $\{w_f\}_f$, the number of iterations $T$.

$\quad \mathbf{Q}_r(\mathbf{v}_r) \leftarrow \frac{1}{\mathbf{z}_r} \exp(\Phi_u(\mathbf{v}_r)))$ for all predicates $r$.

**for** $t \in \{1, ..., T\}$ **do** $\qquad\qquad\qquad \triangleright$ Iterations
$\quad$ **for** $f \in F$ **do** $\qquad\qquad\qquad \triangleright$ formulas
$\quad\quad$ **for** $h \in f$ **do** $\qquad\qquad \triangleright$ Implications
$\quad\quad\quad$ Obtain $\check{\mathbf{Q}}_{r_h}^{[f,h]}(\mathbf{v}_{r_h})$ by Eq. 4. $\triangleright$ Parallel
$\quad\quad$ **end for**
$\quad$ **end for**
$\quad$ Update $\mathbf{Q}_r(\mathbf{v}_r)$ by Eq. 5 for all predicates $r$.
**end for**
**return** $\{\mathbf{Q}_r(\mathbf{v}_r)\}_r$.

---

### 3.3 MULTIPLE MF ITERATIONS AS LOGICMP

Algorithm 1 presents LogicMP, which requires the grouped unary potentials $\{\Phi_u(\mathbf{v}_r)\}_r$, formulas $\{f(\mathcal{A}; \mathbf{n})\}_f$, and rule weights $\{w_f\}_f$, and outputs updated marginals $\{\mathbf{Q}_r(\mathbf{v}_r)\}_r$ for all predicates. LogicMP performs $T$ iterations and firstly computes an initial distribution for every variable using unary potentials and softmax normalization at each iteration. Then, it enumerates all implications to perform Einsum via Eq. 4. The unary potentials and outputs of Einsum are combined to obtain a new estimate of the marginals (Eq. 5). Finally, it returns the output from the last iteration. An execution example is shown in App. I.

## 4 RELATED WORK

**MLN inference.** MLNs are suitable for FOLCs, but have been absent in the neuro-symbolic field for a long time due to inference inefficiency. The most relevant work is the ExpressGNN (Qu & Tang, 2019; Zhang et al., 2020), which has preliminary attempted to combine MLNs with NNs via variational EM. Although both ExpressGNN and LogicMP are based on variational inference, they have clear differences: (1) LogicMP uses the MF algorithm, which permits closed-form iterations (Sec. 2.1). (2) LogicMP obtains essential acceleration by exploiting the structure and symmetries in MLNs (Sec. 3). (3) These enable LogicMP to be applied in general tasks, including computer vision (CV) and natural language processing (NLP) (Sec. 5). Conventional MLN inference methods perform inference either at the level of propositional logic or in a lifted way without performing grounding. The former is inefficient due to the complicated handling of the propositional graph, e.g., Gibbs sampling (Richardson & Domingos, 2006), MC-SAT (Poon & Domingos, 2006), BP (Yedidia et al., 2000). The latter consists of symmetric lifted algorithms which become inefficient with distinctive evidence, such as lifted BP (Singla & Domingos, 2008), and asymmetric lifted algorithms which often requires specific formulas (den Broeck & Davis, 2012; Gribkoff et al., 2014) or evidence (Bui et al., 2012). LogicMP situates itself within the MLN community by contributing a novel and efficient MLN inference method. Dasaratha et al. (2023); Marra et al. (2020) attempted to combine neural networks with first-order logic but the training remains separate.

**Neuro-symbolic reasoning.** A branch of neuro-symbolic methods aims to represent the logic into neural networks (Dasaratha et al., 2023). Besides, some neuro-symbolic methods (Hoernle et al., 2022; Giunchiglia & Lukasiewicz, 2021; Li & Srikumar, 2019; Hoernle et al., 2022; Giunchiglia & Lukasiewicz, 2021; Li & Srikumar, 2019; Fischer et al., 2019; Yang et al., 2022) integrate logical constraints by handling propositional groundings individually. Other methods, e.g., semantic loss (SL) (Xu et al., 2018) and semantic probabilistic layer (SPL) (Ahmed et al., 2022), are rooted in probabilistic logic programming (PLP) that utilizes ACs. However, ACs are often limited to propositional logic and may be insufficient to handle FOLCs unless specific formulas are employed (den Broeck & Davis, 2012). Research applying ACs for FOLCs is currently ongoing in both the MLN and PLP fields, including probabilistic database (Jha & Suciu, 2012) and asymmetric lifted inference (den Broeck &

Niepert, 2015), but it remains a challenging problem. LogicMP exploits the calculation symmetries in MLN for efficient computation by parallel tensor operations. Consequently, LogicMP contributes to developing neuro-symbolic methods for FOLCs using MLNs. Notably, popular neuro-symbolic methods such as DeepProbLog (Manhaeve et al., 2018) and Scallop (Huang et al., 2021) also use ACs and are typically used under closed world assumption rather than OWA.

# 5 EXPERIMENTS

## 5.1 ENCODING FOLC OVER DOCUMENT IMAGES

**Benchmark Dataset.** We apply LogicMP in a CV task, i.e., the information extraction task on the widely used FUNSD form understanding dataset (Jaume et al., 2019). The task involves extracting information from a visual document image, as shown in Fig. 1a, where the model needs to segment tokens into several blocks. The maximum number of tokens is larger than 512. The evaluation metric is the F1 score. The dataset details and general settings are provided in App. J.1.

**Our Method.** We formalize this task as matrix prediction as in (Xu et al., 2022). Each element in the matrix is a binary variable representing whether the corresponding two tokens coexist in the same block. A matrix with ground truth is shown in Fig. 1b. We adopt the LayoutLM (Xu et al., 2020), a robust pre-trained Transformer, as the backbone to derive the vector representation of each token. The matrix $\Phi_u$ is predicted by dot-multiplying each pair of token vectors. We call this model *LayoutLM-Pair*. Independent classifiers often yield unstructured predictions (Fig. 1c), but we can constrain the output via the transitivity of the coexistence, i.e., tokens $a, c$ must coexist when tokens $a, b$ coexist, and $b, c$ coexist. Formally, we denote the predicate as C and the FOLC as $\forall a, b, c : C(a, b) \land C(b, c) \implies C(a, c)$. LogicMP applies this FOLC to LayoutLM-Pair. Each experiment is performed 8 times, and the average score is reported. See App. J.2 for more details.

**Compared Methods.** We compare LogicMP to several robust information extraction methods, including *BIOES* (Xu et al., 2020), *SPADE* (Hwang et al., 2021), and *SpanNER* (Fu et al., 2021). We also compare LogicMP to other neuro-symbolic techniques. *SL* (Xu et al., 2018) is the abbreviation of Semantic Loss, which enforces constraints on predictions by compiling an AC and using it to compute a loss that penalizes joint predictions violating constraints. However, compiling the AC for all variables (up to 262K) is infeasible. Therefore, we use an unrigorous relaxation (*SLrelax*), i.e., penalizing every triplet and summing them via the parallel method proposed in Sec. 3.2. *SPL* (Ahmed et al., 2022) models joint distribution using ACs, but the same relaxation as SL cannot be applied since all variables must be jointly modeled in SPL.

**Main Results.** Table 2 shows the results on the FUNSD dataset, where "full" incorporates all blocks, and "long" excludes blocks with fewer than 20 tokens. Upon integrating FOLC using LogicMP, we observe consistent improvements in two metrics, particularly a 7.3% relative increase in "long" matches. This is because FOLC leverages other predictions to revise low-confidence predictions for distant pairs, as shown in Fig. 1. However, SL and SPL both fail in this task. While attempting to ground the FOLC and compiling AC using PySDD (Darwiche, 2011), we found that it fails when the sequence length exceeds 8 (App. J.3). In contrast, LogicMP can perform joint inference within 0.03 seconds using just 3 tensor operations (App. J.4) with a single additional parameter. SLrelax is beneficial but is outperformed by LogicMP. Additionally, LogicMP is compatible with SLrelax since LogicMP is a neural layer and SLrelax is a learning method with logical regularization. Combining them further improves performance. More visualizations are attached in App. J.5.

## 5.2 ENCODING FOLCS OVER RELATIONAL GRAPHS

**Benchmark Datasets.** We evaluate LogicMP on four collective classification benchmark datasets, each with specific FOLCs. Smoke (Badreddine et al., 2022) serves as a sanity check (see results in App. K.5). Kinship (Zhang et al., 2020) involves determining relationships between people. UW-CSE (Richardson & Domingos, 2006) contains information about students and professors in the CSE department of UW. Cora (Singla & Domingos, 2005) involves de-duplicating entities using the citations between academic papers. It is noteworthy that Cora has 140+K mutually dependent variables and 300+B groundings, with only around 10K known facts. The dataset details and general settings are given in App. K. We conduct each experiment 5 times and report the average results.

Table 2: Comparison of F1 on FUNSD. Better results are in bold. "full" denotes the full set while "long" only considers the blocks with more than 20 tokens. "-" means failure.

| Methods | full | long |
|---|---|---|
| LayoutLM-BIOES (Xu et al., 2020) | 80.1 | 33.7 |
| LayoutLM-SpanNER (Fu et al., 2021) | 74.0 | 22.0 |
| LayoutLM-SPADE (Hwang et al., 2021) | 80.1 | 43.5 |
| LayoutLM-Pair (Xu et al., 2022) | 82.0 | 46.7 |
| LayoutLM-Pair w/ SL (Xu et al., 2018) | - | - |
| LayoutLM-Pair w/ SPL (Ahmed et al., 2022) | - | - |
| LayoutLM-Pair w/ SLrelax | 82.0 | 47.8 |
| LayoutLM-Pair w/ LogicMP | **83.3** | **50.1** |
| LayoutLM-Pair w/ SLrelax+LogicMP | **83.4** | **50.3** |

Table 3: Comparison of F1 on CoNLL2003. Better results are in bold. adj (list) denotes the adjacent (list) rules. "-" means failure.

| Methods | F1 |
|---|---|
| BLSTM (Hu et al., 2016) | 89.98 |
| BLSTM (lex) (Chiu & Nichols, 2016) | 90.77 |
| BLSTM w/ CRF (Lample et al., 2016) | 90.94 |
| BLSTM w/ CRF (mean field) (Wang et al., 2020) | 91.07 |
| BLSTM w/ SL (Xu et al., 2018) | - |
| BLSTM w/ SPL (Ahmed et al., 2022) | - |
| BLSTM w/ SLrelax | 90.38 |
| BLSTM w/ LogicDist (adj) (Hu et al., 2016) | p: 89.80, q: 91.11 |
| BLSTM w/ LogicDist (adj+list) (Hu et al., 2016) | p: 89.93, q: 91.18 |
| BLSTM w/ LogicMP (adj) | 91.25 |
| BLSTM w/ LogicMP (adj+list) | **91.42** |

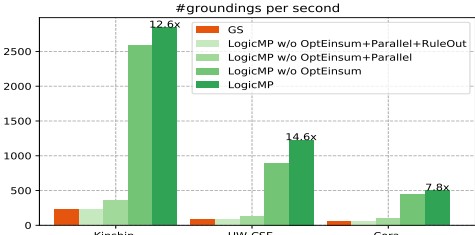

Figure 4: Efficiency comparison of acceleration techniques.

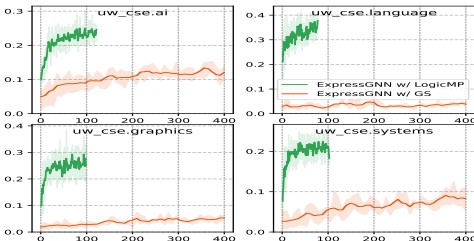

Figure 5: AUC-PR w.r.t minutes during the training in UW-CSE.

**Compared Methods.** We compare with several strong MLN inference methods. *MCMC* (Gilks et al., 1995; Richardson & Domingos, 2006) performs samplings over the ground Markov network. *BP* (Yedidia et al., 2000) uses belief propagation instead. *Lifted BP* (Singla & Domingos, 2008) groups the ground atoms in the Markov network. *MC-SAT* (Poon & Domingos, 2006) performs sampling using boolean satisfiability techniques. *HL-MRF* (Bach et al., 2017; Srinivasan et al., 2019) is hinge-loss Markov random field. *ExpressGNN* denotes the graph neural network proposed in (Zhang et al., 2020), which is trained to fit the data. *ExpressGNN w/ GS* denotes that ExpressGNN is trained to maximize the grounding scores, i.e., the satisfaction of formulas for the groundings using sequential summation (i.e., ExpressGNN-E (Zhang et al., 2020)). Following ExpressGNN w/ GS, we adopt the OWA setting where all unobserved facts are latent variables to infer and use the area under the precision-recall curve (AUC-PR) as the performance evaluation metric.

**Our Method.** For a fair comparison with ExpressGNN w/ GS, we set the rule weights to 1 and use ExpressGNN as the encoding network to obtain unary potentials $\phi_u$. We stack LogicMP with 5 iterations over it. The encoding network is trained to approach the output of LogicMP, which regularizes the output of the encoding network with FOLCs. This learning approach is similar to ExpressGNN w/ GS, as discussed in App. K.4. The main advantage of using LogicMP is its computational efficiency, which enables larger-scale training for better performance.

**Main Results.** Fig. 4 shows the training efficiency of LogicMP, which is about 10 times better than ExpressGNN w/ GS, reducing the computational time per grounding to just 1 millisecond. Thus, we can scale the training from the original 16K (Zhang et al., 2020) to 20M groundings in a reasonable time. Surprisingly, we found that the performance of LogicMP steadily improved with more training (Fig. 5). This observation suggests that the performance of ExpressGNN w/ GS reported in the original work may be hindered by its inefficiency in performing sufficient training.

Table 4 shows the AUC-PR results for the three datasets with a mean standard deviation of 0.03 for UW-CSE and 0.01 for Cora. A hyphen in the entry indicates that it is either out of memory or exceeds the time limit (24 hours). Note that since the lifted BP is guaranteed to get identical results as BP, the results of these two methods are merged into one row. LogicMP obtains almost perfect results on a small dataset (i.e., Kinship), exhibiting its excellent ability in precise inference. In addition, it performs much better than advanced methods on two relatively large datasets (i.e., UW-CSE and

Table 4: AUC-PR on Kinship, UW-CSE, and Cora. The best results are in bold. "-" means failure.

| Method | | Kinship | | | | | | UW-CSE | | | | | | Cora | | | | | |
|---|---|---|---|---|---|---|---|---|---|---|---|---|---|---|---|---|---|---|---|
| | | S1 | S2 | S3 | S4 | S5 | avg. | A. | G. | L. | S. | T. | avg. | S1 | S2 | S3 | S4 | S5 | avg. |
| MLN | MCMC (Richardson & Domingos, 2006) | .53 | - | - | - | - | - | - | - | - | - | - | - | - | - | - | - | - | - |
| | BP/Lifted BP (Singla & Domingos, 2008) | .53 | .58 | .55 | .55 | .56 | .56 | .01 | .01 | .01 | .01 | .01 | .01 | - | - | - | - | - | - |
| | MC-SAT (Poon & Domingos, 2006) | .54 | .60 | .55 | .55 | - | - | .03 | .05 | .06 | .02 | .02 | .04 | - | - | - | - | - | - |
| | HL-MRF (Bach et al., 2017) | **1.0** | **1.0** | **1.0** | **1.0** | - | - | .06 | .09 | .02 | .04 | .03 | .05 | - | - | - | - | - | - |
| NN+ | ExpressGNN | .56 | .55 | .49 | .53 | .55 | .54 | .01 | .01 | .01 | .01 | .01 | .01 | .37 | .66 | .21 | .42 | .55 | .44 |
| | ExpressGNN w/ GS (Zhang et al., 2020) | .97 | .97 | .99 | .99 | .99 | .98 | .09 | .19 | .14 | .06 | .09 | .11 | .62 | .79 | .46 | .57 | .75 | .64 |
| | ExpressGNN w/ LogicMP | .99 | .98 | **1.0** | **1.0** | **1.0** | **.99** | **.26** | **.30** | **.42** | **.25** | **.28** | **.30** | **.80** | **.88** | **.72** | **.83** | **.89** | **.82** |

Cora), improving relatively by 173%/28% over ExpressGNN w/ GS. The improvement is due to its high efficiency, which permits more training within a shorter time (less than 2 hours). Without LogicMP, ExpressGNN w/ GS would take over 24 hours to consume 20M groundings.

**Ablation Study.** Fig. 4 also illustrates the efficiency ablation of the techniques discussed in Sec. 3. As compared to LogicMP, the parallel Einsum technique (Sec. 3.2) achieves significant acceleration, while other improvements, i.e., Einsum optimization and RuleOut (Sec. 3.1), also enhance efficiency. More comparison results are shown in App. K.5.

## 5.3 ENCODING FOLCs OVER TEXT

**Benchmark Dataset & Compared Methods.** We further verify LogicMP in an NLP task, i.e., the sequence labeling task. We conduct experiments on the well-established CoNLL-2003 benchmark (Sang & Meulder, 2003). The task assigns a named entity tag to each word, such as B-LOC, where B is Beginning out of BIOES and LOC stands for "location" out of 4 entity categories. This experiment aims not to achieve state-of-the-art performance but to show that specific FOLCs can also be applied. The compared methods use the bi-directional LSTM (BLSTM) as the backbone and employ different techniques, including *SLrelax* and logic distillation (*LogicDist*) (Hu et al., 2016).

**Our Method.** For a fair comparison, we also use BLSTM as the backbone and stack LogicMP on BLSTM to integrate the following FOLCs used in LogicDist. (1) **adjacent rule**: The BIOES schema contains constraints for adjacent labels, e.g., the successive label of B-PER cannot be O-PER. We explicitly declare the constraints as several adjacent logic rules, such as $\forall i : \texttt{label}(i) \in \{\text{B/I-PER}\} \Leftrightarrow \texttt{label}(i+1) \in \{\text{I/E-PER}\}$, where $\texttt{label}(i)$ is the multi-class predicate of $i$-th token label (see the extension for multi-class predicate in App. E). (2) **list rule**: we exploit a task-specific rule to inject prior knowledge from experts. Specifically, named entities in a list are likely to be in the same categories, e.g., "Barcelona" and "Juventus" in "1. Juventus, 2. Barcelona, 3. ...". The corresponding FOLC is $\forall i, j : \texttt{label}(i) \in \{\text{B/I/E-LOC}\} \land \texttt{samelist}(i, j) \Leftrightarrow \texttt{label}(j) \in \{\text{B/I/E-LOC}\}$, where $\texttt{samelist}(i, j)$ indicates the coexistence of two tokens in a list.

**Main Results.** Table 3 presents our experimental results, with the rule-based methods listed at the bottom. Along with the BLSTM baselines, LogicMP outperforms SLrelax and LogicDist, where "p" denotes BLSTM and "q" post-regularizes the output of BLSTM. These methods implicitly impose constraints during training, which push the decision boundary away from logically invalid prediction regions. In contrast, LogicMP always explicitly integrates the FOLCs into the BLSTM output. For samples with a list structure, LogicMP improves F1 from 94.68 to 97.41.

## 6 CONCLUSION

We presented a novel neuro-symbolic model LogicMP, an efficient method for MLN inference, principally derived from the MF algorithm. LogicMP can act as a neural layer since the computation is fully paralleled through feed-forward tensor operations. By virtue of MLN, LogicMP is able to integrate FOLCs into any encoding network. The output of LogicMP is the (nearly) optimal combination of the FOLCs from MLN and the evidence from the encoding network. The experimental results over various fields prove the efficiency and effectiveness of LogicMP.

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

Table 5: The used symbols and the corresponding denotations.

| Symbol | Definition |
|---|---|
| $r$ | the predicate |
| $f$ | the formula |
| $\|f\|$ | the number of atoms in the formula $f$ |
| $\mathcal{A}^f$ | the arguments of formula $f$ |
| $\|\mathcal{A}^f\|$ | the arity of formula $f$ |
| $g$ | the ground formula (grounding) |
| $O$ | the set of observed facts |
| $i$ | the single ground atom |
| $v_i$ | the single variable to denote the status of ground atom $i$ |
| $\mathbf{v}_g$ | the set of variables w.r.t. ground atoms in the grounding $g$ |
| $G_f$ | the set of groundings of formula $f$ |
| $G_f(i)$ | the set of groundings of formula $f$ that contains $i$ |
| $\phi_u(v_i)$ | the independent unary potential of $i$ with status $v_i$ |
| $\Phi_u(\mathbf{v}_r)$ | the collection of unary potentials w.r.t. $r$ |
| $\phi_f(\mathbf{v}_g)$ | the potential of formula $f$ for $v_g$ |
| $w_f$ | the weight of formula $f$ |
| $\{w_f\}_f$ | the collection of formula weights |
| $\mathbf{n}^f$ | the set of negations of ground atoms in the grounding $g$ w.r.t. $f$ |
| $Q_i$ | the marginal distribution of variable $i$ |
| $\hat{Q}_{i,g}$ | the grounding message of grounding $g$ to the variable $i$ |
| $g_{-i}$ | the set of ground atoms in the grounding $g$ except $i$ |
| $\mathbf{v}_{g_{-i}}$ | the set of variables in the grounding $g$ except $i$ |
| $\mathbf{Q}_r$ | the collection of marginals of predicate $r$ |
| $[f, h]$ | the implication of formula $f$ with the atom $h$ being the hypothesis |
| $\check{\mathbf{Q}}_{r_h}^{[f,h]}$ | the summation of grounding messages w.r.t. the implication $[f, h]$ |

## A  MEAN-FIELD ITERATION EQUATION OF MLN INFERENCE

*Proof.* The conditional probability distribution of MLN in the inference problem is defined as:

$$p(\mathbf{v}|O) \propto \exp\left(\sum_i \phi_u(v_i) + \sum_{f \in F} w_f \sum_{g \in G_f} \phi_f(\mathbf{v}_g)\right). \tag{6}$$

$p(\mathbf{v}|O)$ is generally intractable since there is an exponential summation in the denominator. Therefore, we propose to use a proxy variational distribution $Q(\mathbf{v})$ to approximate the $p(\mathbf{v}|O)$ by minimizing the KL divergence $D_{KL}(Q(\mathbf{v})||p(\mathbf{v}|O))$. The proposed $Q(\mathbf{v})$ is an independent distribution over each variable, i.e., $Q(\mathbf{v}) = \prod_i Q_i(v_i)$ where $\sum_{v_i \in \{0,1\}} Q_i(v_i) = 1, Q_i(v_i) \geq 0$ is a proper probability.

Note that minimizing the KL divergence w.r.t. $Q(\mathbf{v})$ is equivalent to maximizing the evidence lower bound of $\log p(O)$:

$$\begin{aligned} D_{KL}(Q(\mathbf{v})||p(\mathbf{v}|O)) &= \mathbb{E}_{Q(\mathbf{v})} \log \frac{Q(\mathbf{v})}{p(\mathbf{v}|O)} \\ &= \mathbb{E}_{Q(\mathbf{v})} \log Q(\mathbf{v}) - \mathbb{E}_{Q(\mathbf{v})} \log p(\mathbf{v}|O) \\ &= \mathbb{E}_{Q(\mathbf{v})} \log Q(\mathbf{v}) - \mathbb{E}_{Q(\mathbf{v})} \log p(\mathbf{v}, O) + \log p(O) \\ &= -(\mathbb{E}_{Q(\mathbf{v})} \log p(\mathbf{v}, O) - \mathbb{E}_{Q(\mathbf{v})} \log Q(\mathbf{v})) + \log p(O), \end{aligned} \tag{7}$$

which is the negative evidence lower bound (ELBO) plus the log marginal probability of $O$ which is independent of $Q$.

Since $Q(\mathbf{v}) = \prod_i Q_i(v_i)$, we have

$$\mathbb{E}_{Q(\mathbf{v})} \log Q(\mathbf{v}) = \sum_i \sum_{v_i} Q_i(v_i) \log Q_i(v_i). \tag{8}$$

and

$$\mathbb{E}_{Q(\mathbf{v})} \log p(\mathbf{v}|O) = \sum_{i,v_i} \phi_u(v_i) Q_i(v_i) + \sum_{f \in F} w_f \sum_{g \in G_f} \sum_{\mathbf{v}_g} \phi_f(\mathbf{v}_g) \prod_{i \in g} Q_i(v_i) - \log Z, \tag{9}$$

where $Z$ is independent of $Q$.

We can therefore rewrite $D_{KL}(Q(\mathbf{v})||p(\mathbf{v}|O))$ with these two equations as

$$\mathcal{L} = \sum_{i,v_i} Q_i(v_i) \log Q_i(v_i) - \sum_{i,v_i} \phi_u(v_i) Q_i(v_i) - \sum_{f \in F} w_f \sum_{g \in G_f} \sum_{\mathbf{v}_g} \phi_f(\mathbf{v}_g) \prod_{i \in g} Q_i(v_i) + \log Z. \tag{10}$$

Considering it as a function of $Q_i(v_i)$ and remove the irrelevant terms, we have

$$\begin{aligned}\mathcal{L}_i = &\sum_{v_i} Q_i(v_i) \log Q_i(v_i) \\ &- \sum_{v_i} Q_i(v_i)[\phi_u(v_i) + \sum_{f \in F} w_f \sum_{g \in G_f(i)} \sum_{\mathbf{v}_{g-i}} \phi_f(v_i, \mathbf{v}_{g-i}) \prod_{j \in g_{-i}} Q_j(v_i)],\end{aligned} \tag{11}$$

where $g_{-i}$ is the ground variables except $i$ in the grounding $g$, $G_f(i)$ is the groundings of formula $f$ that involve ground atom $i$,

By the Lagrange multiplication theorem with the constraint that $\sum_{v_i} Q_i(v_i) = 1$, the problem becomes

$$\begin{aligned}\arg\min_{Q_i(v_i)} \mathcal{L}_i' = &\sum_{v_i} Q_i(v_i) \log Q_i(v_i) \\ &- \sum_{v_i} Q_i(v_i)[\phi_u(v_i) + \sum_{f \in F} w_f \sum_{g \in G_f(i)} \sum_{\mathbf{v}_{g-i}} \phi_f(v_i, \mathbf{v}_{g-i}) \prod_{j \in g_{-i}} Q_j(v_i)] \\ &+ \lambda(\sum_{v_i} Q_i(v_i) - 1).\end{aligned} \tag{12}$$

Taking the derivative with respect to $Q_i(v_i)$, we have

$$\frac{d\mathcal{L}_i'}{dQ_i(v_i)} = 1 + \log Q_i(v_i) - [\phi_u(v_i) + \sum_{f \in F} w_f \sum_{g \in G_f(i)} \sum_{\mathbf{v}_{g-i}} \phi_f(v_i, \mathbf{v}_{g-i}) \prod_{j \in g_{-i}} Q_j(v_i)] + \lambda. \tag{13}$$

Let the gradient be equal to 0, we then have

$$Q_i(v_i) = \exp(\phi_u(v_i) + \sum_{f \in F} w_f \sum_{g \in G_f(i)} \sum_{\mathbf{v}_{g-i}} \phi_f(v_i, \mathbf{v}_{g-i}) \prod_{j \in g_{-i}} Q_j(v_i) - 1 - \lambda). \tag{14}$$

Taking $\lambda$ out from the equation, we have

$$Q_i(v_i) = \frac{1}{Z_i} \exp(\phi_u(v_i) + \sum_{f \in F} w_f \sum_{g \in G_f(i)} \sum_{\mathbf{v}_{g-i}} \phi_f(v_i, \mathbf{v}_{g-i}) \prod_{j \in g_{-i}} Q_j(v_j)), \tag{15}$$

where $Z_i$ is the partition function.

For clarity of presentation, we define the message of a single grounding (grounding message) as

$$\begin{aligned}Q_i(v_i) &= \frac{1}{Z_i} \exp(\phi_u(v_i) + \sum_{f \in F} w_f \sum_{g \in G_f(i)} \hat{Q}_{i,g}(v_i)), \\ \hat{Q}_{i,g}(v_i) &= \sum_{\mathbf{v}_{g-i}} \phi_f(v_i, \mathbf{v}_{g-i}) \prod_{j \in g_{-i}} Q_j(v_j).\end{aligned} \tag{16}$$

Then we have the conclusion. $\qquad\qquad\qquad\qquad\qquad\qquad\qquad\qquad\qquad\qquad\qquad\qquad\square$

# B Terminology

**Literal.** In mathematical logic, a literal is an atomic formula (also known as an atom or prime formula) or its negation. Literals can be divided into two types: a positive literal is just an atom (e.g., $l$) and a negative literal is the negation of an atom (e.g., $\neg l$).

**Clause.** A clause is a formula formed from a finite disjunction of literals. A clause is true whenever at least one of the literals that form it is true. Clauses are usually written as $(l_1 \vee l_2...)$, where the symbols $l_i$ are literals, e.g., $\texttt{S}(a) \wedge \texttt{F}(a, b) \implies \texttt{S}(b)$.

**Implication.** Every nonempty clause is logically equivalent to an implication of a head from a body, where the head is an arbitrary literal of the clause and the body is the conjunction of the negations of the other literals. The equivalent implications of $\texttt{S}(a) \wedge \texttt{F}(a, b) \implies \texttt{S}(b)$ are (1) $\texttt{S}(a) \wedge \texttt{F}(a, b) \implies \texttt{S}(b)$, (2) $\texttt{S}(a) \wedge \neg\texttt{S}(b) \implies \neg\texttt{F}(a, b)$ and (3) $\texttt{F}(a, b) \wedge \neg\texttt{S}(b) \implies \neg\texttt{S}(a)$

**Conjunctive Normal Form (CNF).** A formula is in conjunctive normal form if it is a conjunction of one or more clauses, where a clause is a disjunction of literals. The CNF is usually written as $(l_{1,1} \vee l_{1,2}...) \wedge (l_{2,1} \vee l_{2,2}...) \wedge ...$, where the symbols $l_{i,j}$ are literals, e.g., $(\texttt{S}(a) \implies \texttt{C}(a)) \wedge (\texttt{C}(a) \implies \texttt{S}(a))$.

**First-order Logic (FOL).** FOL uses quantified variables over a set of constants, and allows the use of sentences that contain variables, so that rather than propositions such as "Tom is a father, hence Tom is a man", one can have expressions in the form "for all person $x$, if $x$ is a father then $x$ is a man", where "for all" is a quantifier, while $x$ is an argument. In general, the clause and CNF are used for propositional logic which does not use quantifiers. In this work, we denote the first-order clause and first-order CNF with universal quantifiers, respectively.

**Ground Expression.** A ground term of a formal system is a term that does not contain any variables. Similarly, a ground formula is a formula that does not contain any variables. A ground expression is a ground term or ground formula. In this paper, the grounding of a formula means assigning values to the variables in the formula with universal quantifiers.

# C Proof of Theorem: Message of Clause Considers True Premise Only

**Lemma C.1.** *(No message of clause for the false premise.) When each formula $f(\cdot; \mathbf{n})$ is a clause, for a particular state $\mathbf{v}_{g_{-i}}^*$ of a grounding $g \in G_f(i)$ that $\exists j \in g_{-i}, v_j^* = \neg n_j$, the MF iteration of Eq. 2 is equivalent for $\hat{Q}_{i,g}(v_i) \leftarrow \sum_{\mathbf{v}_{g_{-i}} \neq \mathbf{v}_{g_{-i}}^*} \phi_f(v_i, \mathbf{v}_{g_{-i}}) \prod_{j \in g_{-i}} Q_j(v_j)$.*

*Proof.* Let us consider a grounding $g^*$ in $G_f(i)$ and a grounding message from $g_{-i}^*$ to $i$ and there is a particular state $\mathbf{v}_{g_{-i}^*}^*$ that $\exists j \in g_{-i}^*, v_j^* = \neg n_j$. We explicitly extract the grounding message of state $\mathbf{v}_{g_{-i}^*}^*$ from the overall potential as below,

$$
\begin{aligned}
Q_i(v_i) &= \frac{\exp(E_i(v_i))}{\sum_{v_i} \exp(E_i(v_i))}, \\
E_i(v_i) &= \phi_f(v_i, \mathbf{v}_{g_{-i}^*}^*; \mathbf{n}) \prod_{j \in g_{-i}^*} Q_j(v_j^*) + \Delta_i(v_i), \\
\Delta_i(v_i) &= \phi_u(v_i) + \sum_{\mathbf{v}_{g_{-i}^*} \neq \mathbf{v}_{g_{-i}^*}^*} \phi_f(v_i, \mathbf{v}_{g_{-i}^*}; \mathbf{n}) \prod_{j \in g_{-i}^*} Q_j(v_j) \\
&\quad + \sum_f w_f \sum_{g \in G_f(i) \backslash g^*} \sum_{\mathbf{v}_{g_{-i}}} \phi_f(v_i, \mathbf{v}_{g_{-i}}; \mathbf{n}^f) \prod_{j \in g_{-i}} Q_j(v_j)).
\end{aligned}
\tag{17}
$$

Since $\exists j \in g_{-i}^*, v_j^* = \neg n_j^f$, the clause will always be true regardless of $v_i$, i.e., $\forall v_i \in \{0, 1\}$, $\phi_f(v_i, \mathbf{v}_{g_{-i}^*}^*) = 1$.

Therefore, grounding message of state $\mathbf{v}^*_{g^*_{-i}}$ (i.e., $\phi_f(v_i, \mathbf{v}^*_{g^*_{-i}}; \mathbf{n}^f) \prod_{j \in g^*_{-i}} Q_j(v^*_j)$) is independent of $v_i$, then

$$E_i(v_i) = C + \Delta_i(v_i), C = w_f \prod_{j \in g^*_{-i}} Q_j(v^*_j). \tag{18}$$

The potential of $C$ is eliminated in the normalization step for $v_i = 0$ and $v_i = 1$. We can apply the same logic to all the grounding messages and obtain the conclusion: $Q_i(v_i) = \frac{1}{Z_i} \exp(\phi_u(v_i) + \sum_f w_f \sum_{g \in G_f(i)} \hat{Q}_{i,g}(v_i))$, where $\hat{Q}_{i,g}(v_i) = \sum_{\mathbf{v}_{g-i} \neq \mathbf{v}^*_{g-i}} \phi_f(v_i, \mathbf{v}_{g-i}) \prod_{j \in g_{-i}} Q_j(v_j)$. □

The proof of the theorem is simple by using this lemma.

*Proof.* By the lemma C.1, only one remaining state needs to be considered, i.e., $\{v_j = n_j\}_{j \in g_{-i}}$. The potential $\phi_f(\cdot)$ is 1 iff $v_i = \neg n_i$, otherwise $\hat{Q}_{i,g}(v_i) \leftarrow 0$. Then we derive the conclusion: $Q_i(v_i) \leftarrow \frac{1}{Z_i} \exp(\phi_u(v_i) + \sum_f w_f \sum_{g \in G_f(i)} \hat{Q}_{i,g}(v_i))$, where $\hat{Q}_{i,g}(v_i) \leftarrow \mathbf{1}_{v_i = \neg n^f_i} \prod_{j \in g_{-i}} Q_j(v_j = n^f_j)$.

□

# D   PROOF OF THEOREM: MESSAGE OF CNF = $\sum$ MESSAGE OF CLAUSE

*Proof.* For convenience, let us consider a grounding $g^*$ in $G_f(i)$ where $f$ in CNF is the conjunction of several distinct clauses $f_k(\cdot; \mathbf{n}^{f_k})$:

$$\begin{aligned}
Q_i(v_i) &= \frac{\exp(E_i(v_i))}{\sum_{v_i} \exp(E_i(v_i))}, \\
E_i(v_i) &= \sum_{\mathbf{v}_{g^*_{-i}}} \phi_f(v_i, \mathbf{v}_{g^*_{-i}}) \prod_{j \in g^*_{-i}} Q_j(v_j) + \Delta_i(v_i), \\
\Delta_i(v_i) &= \phi_u(v_i) + \sum_f w_f \sum_{g \in G_f(i) \backslash g^*} \sum_{\mathbf{v}_{g-i}} \phi_f(v_i, \mathbf{v}_{g-i}) \prod_{j \in g^*_{-i}} Q_j(v_j)).
\end{aligned} \tag{19}$$

Let $\mathbf{v}^k_{g^*_{-i}}$ be $\{n^{f_k}_j\}_{j \in g^*_{-i}}$, i.e., the corresponding true premises of clauses. We have

$$E_i(v_i) = \sum_k \phi_f(v_i, \mathbf{v}^k_{g^*_{-i}}) \prod_{j \in \mathbf{v}^k_{g^*_{-i}}} Q_j(v^k_j) + \sum_{\mathbf{v}_{g^*_{-i}} \notin \{\mathbf{v}^k_{g^*_{-i}}\}} \phi_f(v_i, \mathbf{v}_{g^*_{-i}}) \prod_{j \in g^*_{-i}} Q_j(v_j) + \Delta_i(v_i), \tag{20}$$

where the second term can be directly eliminated as in the proof of Lemma C.1. We consider two cases that $\mathbf{v}^k_{g^*_{-i}}$ is unique or not for various $k$.

- Case 1: $\mathbf{v}^k_{g^*_{-i}}$ is unique: Since $\mathbf{v}^k_{g^*_{-i}}$ is unique, then $\phi_f(v_i, \mathbf{v}^k_{g^*_{-i}}) = \mathbf{1}_{v_i = \neg n^{f_k}_i}$ and we can get the message by the same logic in Theorem 3.1, i.e., $\mathbf{1}_{v_i = \neg n^{f_k}_i} \prod_{j \in g^*_{-i}} Q_j(v_j = n^{f_k}_j)$.

- Case 2: $\mathbf{v}^k_{g^*_{-i}}$ is not unique: Let $\mathbf{v}^{k_1}_{g^*_{-i}}$ and $\mathbf{v}^{k_2}_{g^*_{-i}}$ be the same where $k_1$ and $k_2$ are two clauses in $f$. Since the clauses are unique, $n^{f_{k_1}}_i$ must be different with $n^{f_{k_2}}_i$. The two potentials are eliminated, which is equivalent to the summation of distinct messages, i.e., $\sum_{k_1, k_2} \mathbf{1}_{v_i = \neg n^{f_k}_i} \prod_{j \in g^*_{-i}} Q_j(v_j = n^{f_k}_j)$.

We can apply this logic for all possible groundings and obtain:

$$Q_i(v_i) \leftarrow \frac{1}{Z_i} \exp(\phi_u(v_i) + \sum_f w_f \sum_{g \in G_f(i)} \hat{Q}_{i,g}(v_i)),$$

where $\hat{Q}_{i,g}(v_i) \leftarrow \sum_k \mathbf{1}_{v_i = \neg n^{f_k}_i} \prod_{j \in g_{-i}} Q_j(v_j = n^{f_k}_j)$, i.e., $\sum_{f_k} \mathbf{1}_{v_i = \neg n_i} \prod_{j \in g_{-i}} Q_j(v_j = n_j)$.

□

This theorem directly leads to the following corollary.

**Corollary D.1.** *For the MLN, the mean-field update w.r.t. a CNF formula is equivalent to the mean-field update w.r.t. multiple clause formulas with the same rule weight.*

## E EXTENSION OF MULTI-CLASS PREDICATES

A typical MLN is defined over binary variables where the corresponding fact can be either true or false. This is unusual in the modeling of common tasks, where the exclusive categories form a single multi-class classification. For instance, a typical MLN will use several binary predicates to describe the category of a paper by checking whether the paper is of a certain category. However, the categories are typically exclusive, and combining them can ease model learning. Therefore, we extend LogicMP to model multi-class predicates.

Formally, let the predicates be multi-class classifications $r(\cdot) : C \times ... \times C \rightarrow \{0, 1, ...\}$ with $\geq 2$ categories, which is different with standard MLN. The atom in the formula is then equipped with another configuration $\mathcal{Z}$ for the valid value of predicates. For instance, a multi-class formula about "RL paper cites RL paper" can be expressed as

$$\texttt{label}(x) \in \{RL\} \vee \texttt{cite}(x, y) \implies \texttt{label}(y) \in \{RL\}, \tag{21}$$

where $\texttt{label}(x) \in RL$ means $x$ is a RL paper. Sometimes the predicates in the formula appear more than one time, e.g., $\texttt{label}(x) \in \{ML\} \vee \texttt{label}(x) \in \{RL\}$.... We should aggregate them into a single literal $\texttt{label}(x) \in \mathcal{Z}, \mathcal{Z} = \{RL, ML\}$. A clause with multi-class predicates is then formulated as $... \vee (v_i \in \mathcal{Z}_i) \vee ...$ where $v_i$ is the variable associated with the atom $i$ in the clause and $\mathcal{Z}_i$ denotes the possible values the predicate can take. With such notation, we rewrite the clauses with multi-class predicates as $f(\cdot; \mathcal{Z}^f)$ where $\mathcal{Z}^f = \{\mathcal{Z}_i\}_i$. We show that the message of the multi-class clause can be derived by the following theorem.

**Theorem E.1.** *When each formula with multi-class predicates $f(\cdot; \mathcal{Z}^f)$ is a clause, the MF iteration of Eq. 2 is equivalent for $\hat{Q}_i(v_i) = \mathbf{1}_{v_i \in \mathcal{Z}_i} \prod_{j \in g_{-i}} (1 - \sum_{v_j \in \mathcal{Z}_j} Q_j(v_j))$.*

As the derivation is similar to that of binary predicates, we omit the detailed proof here. By setting $\mathcal{Z} = \{\neg n_i\}$ in the binary case, we can see that the message with multi-class predicates becomes the one with binary predicates. Similarly, when the formula is the CNF, the message can be calculated by aggregating the messages of clauses.

## F PARALLEL AGGREGATION USING EINSTEIN SUMMATION

The virtue of parallel computation lies in the summation of the product, i.e., $\sum_{g \in G_f(i)} \prod_{j \in g_{-i}} Q_j(v_j = n_j)$ by Theorem 3.1, which indicates that the grounding message corresponds to the implication from the premise $g_{-i}$ to the hypothesis $i$. By the structure of MLN, many grounding messages belong to the same implication and have the calculation symmetries, so we group them by their corresponding implications. The aggregation of grounding messages w.r.t. an implication amounts to integrating some rule arguments and we can therefore formalize the aggregation into Einsum. For instance, the aggregation w.r.t. the implication $\texttt{S}(a) \wedge \texttt{F}(a, b) \implies \texttt{S}(b)$ can be expressed as $\texttt{einsum}(\text{"}a, ab \rightarrow b\text{"}, \mathbf{Q}_\texttt{S}(\mathbf{1}), \mathbf{Q}_\texttt{F}(\mathbf{1}))$ where $\mathbf{Q}_r(\mathbf{v}_r)$ denotes the collection of marginals w.r.t. predicate $r$, i.e., $\mathbf{Q}_r(\mathbf{v}_r) = \{Q_{r(\mathcal{A}_r)}(v_{r(\mathcal{A}_r)})\}_{\mathcal{A}_r}$ ($\mathcal{A}_r$ is the arguments of $r$). Fig. 6 illustrates this process, where we initially group the variables by predicates and then use them to perform aggregation using parallel tensor operations. The LogicMP layer on the right of Fig. 6 gives a detailed depiction of the mechanism, where 3 clauses generate 7 implications and each first-order implication statement is transformed into an Einsum operation for parallel message computation. Einsum also allows the computation for other complex formulae whose predicates have many arguments.

## G EINSUM EXAMPLES

Einstein summation [1] is the notation for the summation of the product of elements in a list of high-dimensional tensors. We found the aggregation of grounding messages w.r.t. an implication

---

[1] https://en.wikipedia.org/wiki/Einstein_notation

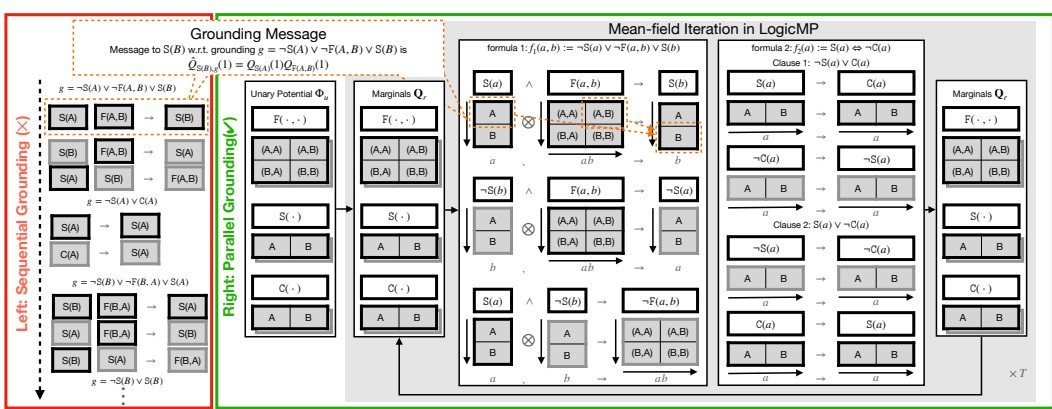

Figure 6: We illustrate the LogicMP for a Markov logic network (MLN) with two entities $\{A, B\}$, predicates F (Friend) and S (Smoke) and C (Cancer), and formulae $f_1(a, b) := \neg S(a) \lor \neg F(a, b) \lor S(b)$, $f_2(a) := (\neg S(a) \lor C(a)) \land (S(a) \lor \neg C(a))$. **Left**: Vanilla mean-field implementation performs sequential grounding. **Right**: We expand the MLN inference into several mean-field iterations and each iteration is implemented by a layer of LogicMP. The red "Grounding Message" block illustrates the message w.r.t. a single grounding: the message from $S(A)$ and $F(A, B)$ w.r.t. $f_1(A, B)$ can be simplified as the product of their marginals $Q_{S(A)}(1)Q_{F(A,B)}(1)$ (see Sec. 3.1). We then formulate message aggregation via the parallel Einstein summation (Einsum)(see Sec. 3.2). Specifically, the ground atoms are grouped by the predicates as the basic units of computation (the gray border denotes the values of $\neg$). The initial marginals are obtained from distinct evidence. In each iteration, the layer of LogicMP takes them as input and performs Einsum for each logic rule (shown in the "LogicMP Layer" block). For each implication statement of the logic rules, Einsum calculates the expected number of the groundings that derive the hypothesis. The outputs of Einsum are then used to update the grouped marginals. Such procedure loops for $T$ steps until convergence. Note that $f_2$ is in CNF and its two clauses can be treated separately. The Einsum is also applicable for predicates with more than two arguments. The update takes several updates to converge.

can be exactly represented by an Einstein summation expression. Notably, Einstein summation can be efficiently implemented in parallel via NumPy and nowadays deep learning frameworks, e.g., PyTorch and TensorFlow. The corresponding function is called `einsum` [2] which can be effectively optimized via a library `opt_einsum` [3]. We list several cases of message aggregation in the Einstein summation format and their optimal simplifications via dynamic argument contraction. Deriving the optimal solution is an NP-hard problem, but for the practical formulas whose argument size (i.e., arity) is less than 6, the solution can be obtained within milliseconds. For all the formulas in the tested datasets, the number of arguments is less than 6.

Formula: $R_1(h, k) \wedge R_2(k, j) \wedge R_3(j, i) \to R_0(i)$

- Original: $\mathbf{K} \leftarrow \texttt{einsum}(\text{``}hk, kj, ji \to i\text{''}, \mathbf{Q}_{R_1}(\mathbf{1}), \mathbf{Q}_{R_2}(\mathbf{1}), \mathbf{Q}_{R_3}(\mathbf{1}))$
- Optimized:
  - $\mathbf{K} \leftarrow \texttt{einsum}(\text{``}kj, ji \to ki\text{''}, \mathbf{Q}_{R_2}(\mathbf{1}), \mathbf{Q}_{R_3}(\mathbf{1}))$
  - $\mathbf{K} \leftarrow \texttt{einsum}(\text{``}hk, ki \to i\text{''}, \mathbf{Q}_{R_1}(\mathbf{1}), \mathbf{K})$

Formula: $R_1(h, k) \wedge R_2(k, j) \wedge R_3(j, i) \wedge R_4(h) \to R_0(i)$

- Original: $\mathbf{K} \leftarrow \texttt{einsum}(\text{``}hk, kj, ji, h \to i\text{''}, \mathbf{Q}_{R_1}(\mathbf{1}), \mathbf{Q}_{R_2}(\mathbf{1}), \mathbf{Q}_{R_3}(\mathbf{1}), \mathbf{Q}_{R_4}(\mathbf{1}))$
- Optimized:
  - $\mathbf{K} \leftarrow \texttt{einsum}(\text{``}kj, ji \to ki\text{''}, \mathbf{Q}_{R_2}(\mathbf{1}), \mathbf{Q}_{R_3}(\mathbf{1}))$
  - $\mathbf{K} \leftarrow \texttt{einsum}(\text{``}hk, h, ki \to i\text{''}, \mathbf{Q}_{R_1}(\mathbf{1}), \mathbf{Q}_{R_4}(\mathbf{1}), \mathbf{K})$

Formula: $R_1(p, i) \wedge R_1(q, j) \wedge R_2(i, j, k, l) \wedge R_1(r, k) \wedge R_1(s, l) \to R_0(p, q, r, s)$

- Original: $\mathbf{K} \leftarrow \texttt{einsum}(\text{``}pi, qj, ijkl, rk, sl \to pqrs\text{''}, \mathbf{Q}_{R_1}(\mathbf{1}), \mathbf{Q}_{R_1}(\mathbf{1}), \mathbf{Q}_{R_2}(\mathbf{1}), \mathbf{Q}_{R_1}(\mathbf{1}), \mathbf{Q}_{G}(\mathbf{1}))$
- Optimized:
  - $\mathbf{K} \leftarrow \texttt{einsum}(\text{``}pi, ijkl \to pjkl\text{''}, \mathbf{Q}_{R_1}(\mathbf{1}), \mathbf{Q}_{R_2}(\mathbf{1}))$
  - $\mathbf{K} \leftarrow \texttt{einsum}(\text{``}qj, pjkl \to pqkl\text{''}, \mathbf{Q}_{R_1}(\mathbf{1}), \mathbf{K})$
  - $\mathbf{K} \leftarrow \texttt{einsum}(\text{``}rk, pqkl \to pqrl\text{''}, \mathbf{Q}_{R_1}(\mathbf{1}), \mathbf{K})$
  - $\mathbf{K} \leftarrow \texttt{einsum}(\text{``}sl, pqrl \to pqrs\text{''}, \mathbf{Q}_{R_1}(\mathbf{1}), \mathbf{K})$

We also show several cases that cannot be optimized, as follows.

Formula: $R_1(a) \wedge R_2(a, b) \to R_1(b)$

- $\mathbf{K} \leftarrow \texttt{einsum}(\text{``}a, ab \to b\text{''}, \mathbf{Q}_{R_1}(\mathbf{1}), \mathbf{Q}_{R_2}(\mathbf{1}))$

Formula: $R_1(a, b, c, d) \wedge R_2(b, c) \wedge R_3(c, d) \wedge R_4(a, d) \to R_0(a, c)$

- $\mathbf{K} \leftarrow \texttt{einsum}(\text{``}abcd, bc, cd, ad \to ac\text{''}, \mathbf{Q}_{R_1}(\mathbf{1}), \mathbf{Q}_{R_2}(\mathbf{1}), \mathbf{Q}_{R_3}(\mathbf{1}), \mathbf{Q}_{R_4}(\mathbf{1}))$

Formula: $R_1(a, b, c) \wedge R_2(b, c, d) \wedge R_3(c, b) \wedge R_4(a, d) \to R_0(a, c)$

- $\mathbf{K} \leftarrow \texttt{einsum}(\text{``}abc, bcd, cb, ad \to ac\text{''}, \mathbf{Q}_{R_1}(\mathbf{1}), \mathbf{Q}_{R_2}(\mathbf{1}), \mathbf{Q}_{R_3}(\mathbf{1}), \mathbf{Q}_{R_4}(\mathbf{1}))$

One may notice that the current implementations of the Einsum function are not available when the target matrix has external arguments that are not in the input matrices, e.g., "$a \to ab$". We tackle this by a post-processing function for the output of Einsum.

## H   COMPLEXITY OF OPTIMIZED EINSTEIN SUMMATION

The optimized Einsum is of complexity $\mathcal{O}(N^{M'})$ where $M'$ is the maximum number of arguments in the sequence of argument contraction operations. The optimization of Einsum, i.e., the summation of the product of a list of tensors, is equivalent to a classical problem in probabilistic graph modeling, i.e., the variable elimination in the summation of the product of the potentials. Each step of Einsum after the optimization contracts the rule arguments in the expression, which amounts to a step of the variable

---

[2] https://pytorch.org/docs/stable/generated/torch.einsum.html
[3] https://optimized-einsum.readthedocs.io/en/stable/

Table 6: A possible world (assignment of variables) of MLN.

| | $v_{\mathtt{S}(A)}$ | $v_{\mathtt{S}(B)}$ | $v_{\mathtt{F}(A,A)}$ | $v_{\mathtt{F}(A,B)}$ | $v_{\mathtt{F}(B,B)}$ | $v_{\mathtt{C}(A)}$ |
|---|---|---|---|---|---|---|
| binary value | 1 | 1 | 0 | 1 | 0 | 1 |

Table 7: An example of variable marginals.

| | $Q_{\mathtt{S}(A)}(1)$ | $Q_{\mathtt{S}(B)}(1)$ | $Q_{\mathtt{F}(A,A)}(1)$ | $Q_{\mathtt{F}(A,B)}(1)$ | $Q_{\mathtt{F}(B,B)}(1)$ | $Q_{\mathtt{C}(A)}(1)$ |
|---|---|---|---|---|---|---|
| continuous value | 0.92 | 0.97 | 0.13 | 0.95 | 0.03 | 0.99 |

elimination in the message passing. Even though, the goal of Einsum in LogicMP, which functions on the level of logical groundings, differs greatly from the original message-passing algorithms, which function on the standard (directed or undirected) probabilistic graphs. By formulating the optimization of Einsum as a variable elimination algorithm, we can see $M'$ also equals the maximum cluster size of the junction tree constructed in the variable elimination. We define the junction tree for Einsum of LogicMP as:

**Definition H.1.** Given the Einsum expression in LogicMP, a junction tree of LogicMP is a tree $\mathcal{T}(V, E)$ in which each vertex $v \in V$ (also called a cluster) and edge $e \in E$ are labeled with a subset of formula arguments, denoted by $L(v)$ and $L(e)$ such that: (i) for every atom $r$ defined in Einsum, there exists a vertex $L(v)$ such that $\mathcal{A}_r \subseteq L(v)$ and (ii) for every argument $x$ in the Einsum expression, the set of vertexes and edges in $\mathcal{T}$ that mention $x$ form a connected sub-tree in $\mathcal{T}$ (called the running intersection property).

A previous study (Venugopal et al., 2015) proposed to count the true groundings in MLN by formulating a constraint satisfaction problem (CSP). In contrast to counting the true groundings, the Einsum calculates the expected number of groundings that derives the hypothesis (i.e., true premise) over the marginal distributions of the ground atoms. It is in principle different from previous methods since our approach is theoretically derived from the mean-field algorithm. More importantly, Einsum is a parallel tensor operation that brings computational efficiency and enables LogicMP to be an effective neural module. By using Einsum, MLN inference can be expressed as an efficient parallel NN.

## I  AN EXAMPLE TO ILLUSTRATE THE LOGICMP ALGORITHM

Let's take the Smoke dataset as an example. The predicates are smoke(x) ($\mathtt{S}(x)$), friend(x, y) ($\mathtt{F}(x,y)$) and cancer(x) ($\mathtt{C}(x)$). The observations $O$ are two facts, i.e., $\mathtt{F}(B,A) = true$ and $\mathtt{C}(B) = true$. The unobserved variables $\mathbf{v}$ are $v_{\mathtt{S}(A)}, v_{\mathtt{S}(B)}, v_{\mathtt{F}(A,A)}, v_{\mathtt{F}(A,B)}, v_{\mathtt{F}(B,B)}, v_{\mathtt{C}(A)}$. Each variable $v_i$, e.g., $v_{\mathtt{S}(A)} \in \{0, 1\}$, is a binary discrete variable. An assignment to these variables corresponds to a world. A possible world in MLN is shown in Table 6.

In our variational algorithm, LogicMP mitigates the problem of counting exponential worlds by updating in the single world of approximate marginal probabilities of ground atoms where einsum is used to calculate the expected number of all possible true premises. Each variable $v_i$ is represented by a marginal probability $Q_i(v_i) \in [0, 1]$, e.g., $Q_{\mathtt{S}(A)}(1) = 0.92$ ($Q_{\mathtt{S}(A)}(0) = 0.08$) denotes that the marginal of probability of $\mathtt{S}(A)$ being true is 0.92 (0.08 for false). The marginals may be as shown in Table 7.

The mean-field update is computed by these 6 marginals via Proposition 3.2, which updates new marginals of variables conditioned on the current marginals of variables. We illustrate how our algorithm updates the marginals of $\mathtt{S}(A)$ and $\mathtt{S}(B)$, i.e., $\mathbf{Q_S}$, as follows. They receive messages from 4 implications statements and we calculate them via einsum as illustrated in Table 8. The updated marginals are:

$$\mathbf{Q_S}(1) = \exp(\Phi_{\mathtt{S}}(1) + w_1 e_1 + w_2 e_3)/\mathbf{Z_S} \, ,$$
$$\mathbf{Q_S}(0) = \exp(\Phi_{\mathtt{S}}(0) + w_1 e_2 + w_2 e_4)/\mathbf{Z_S} \, ,$$

where $\mathbf{Z_S} = \exp(\Phi_{\mathtt{S}}(1) + w_1 e_1 + w_2 e_3) + \exp(\Phi_{\mathtt{S}}(0) + w_1 e_2 + w_2 e_4)$, $w_i$ is the rule weight, all computations are based on the marginal probabilities of ground atoms, i.e, $Q_i$s.

Table 8: Illustration of the Message Calculation. EN indicates the expected number.

| implication | einsum | variational approximation |
|---|---|---|
| $S(x) \wedge F(x,y) \implies S(y)$ | $e_1 =$einsum("$x, xy \to y$", $\mathbf{Q}_S(1), \mathbf{Q}_F(1)$) | EN of the true ground premises $S(x) \wedge F(x,y)$ |
| $\neg S(y) \wedge F(x,y) \implies \neg S(x)$ | $e_2 =$einsum("$y, xy \to x$", $1 - \mathbf{Q}_S(1), \mathbf{Q}_F(1)$) | EN of the true ground premises $\neg S(y) \wedge F(x,y)$ |
| $C(x) \implies S(x)$ | $e_3 =$einsum("$x \to x$", $\mathbf{Q}_C(1)$) | EN of the true ground premises $C(x)$ |
| $\neg C(x) \implies \neg S(x)$ | $e_4 =$einsum("$x \to x$", $1 - \mathbf{Q}_C(1)$) | EN of the true ground premises $\neg C(x)$ |

## J    MORE RESULTS ON VISUAL DOCUMENT UNDERSTANDING

### J.1    GENERAL SETTINGS

The experiments were conducted on a basic machine with a 132GB V100 GPU and an Intel E5-2682 v4 CPU at 2.50GHz with 32GB RAM. The dataset consists of 149 training samples and 50 test samples. A sample consists of an input image and a sequence of tokens with their corresponding 2-D positions. The number of tokens may be larger than 512. The tokens are segmented into several blocks and each block belongs to a category out of "other", "question", "answer", "header". The goal of this task is to segment the tokens into blocks and label each block correctly. Specifically, given the image document with image pixels and tokens from OCR (e.g., an image document with text "Date: February 25, 1998 ... CLIENT TO: L8557.002 ..."), the goal is to segment the tokens into blocks (e.g., "February 25, 1998" should be segmented into one "answer" block). The maximum token count is 512, and each token has an ID ranging from 0 to 511 (e.g., the ID of "February" is 2, and "25" is 3).

### J.2    DETAILS OF OUR METHOD

We formalize this task as a matrix prediction task with reference to previous work (Xu et al., 2022). The model framework is shown in Fig. 7. It takes image pixels and tokens as input, and outputs the predictions about whether pairs of tokens coexist within a block. Let $C$ denote the coexistence predicate and $C(a,b)$ denote whether tokens $a$ and $b$ coexist. The model needs to predict all $\{C(a,b)\}_{(a,b)}$ which forms a matrix. A matrix with ground truth is shown in Fig. 8b.

Specifically, a data sample consists of $n$ input visual token entities $\{e_1, ..., e_N\}$. We adopt the LayoutLM (Xu et al., 2020), which is a powerful pre-trained Transformer with visual and text inputs, as the backbone to derive the vector representation of each token, i.e., $\mathbf{V} \in \mathcal{R}^{N \times d}$ where $N$ is the number of token entities and $d$ is the dimension of vector (768). Then we apply two linear transformation layers to map $\mathbf{V}$ to $\mathbf{V}^r$ and $\mathbf{V}^c$ as rows and columns. The similarity of $\mathbf{V}^r_a$ and $\mathbf{V}^c_b$, captured by their dot multiplication, is used as the unary potential, i.e., $\Phi_u(v_{C(a,b)})$. A data sample also manually annotated with blocks containing tokens. We convert the blocks into a target matrix of coexistence $\mathbf{Y}_c \in \{0,1\}^{N \times N}$.

Independent classifier takes $\Phi_u$ as input and outputs the prediction via a Sigmoid layer over $\Phi_u(v_{C(\cdot,\cdot)})$. The independent classifier often yields conflict predictions (Fig. 8c) but we can constrain the output via the transitivity of the coexistence. Intuitively, when tokens $a$ and $b$ coexist and tokens $b$ and $c$ coexist, tokens $a$ and $c$ must coexist. Formally, we denote the predicate of the matrix as $C$ and the FOLC is $\forall a, b, c : C(a,b) \wedge C(b,c) \implies C(a,c)$. LogicMP applies this FOLC to this matrix prediction to constrain the output. This rule should be effective considering that the distant tokens are not easily predicated but can be inferred by the coexistence of other closer and easier pairs. The entire model is trained to minimize the cross-entropy between the prediction and the labels, using the Adam optimizer with a learning rate of 5e-4.

### J.3    AC COMPILATION

Both SL and SPL rely on the AC but AC compilation will fail in this task. Although specific first-order logic (FOL) rules can be effectively compiled into ACs, such as those liftable queries under the Big Dichotomy theorem (Dalvi & Suciu, 2013), the transitivity rule falls into the category where the compilation is intractable. Specifically, the AC compilation time consumption is shown in Table 9. The table shows that the compilation time is exponential in the sequence length. Once the sequence length reaches 8, the compilation time becomes unreasonably long. In contrast, LogicMP is capable

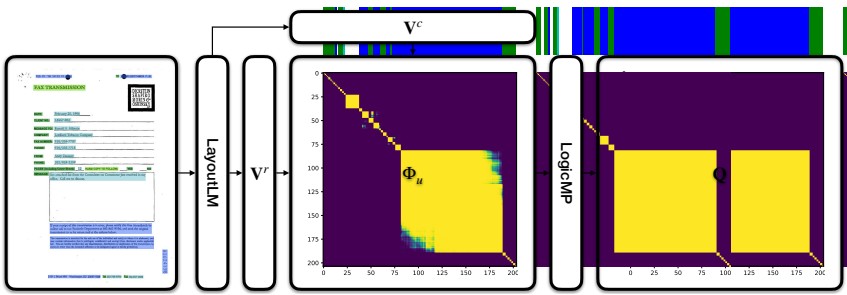

Figure 7: We adopt the LayoutLM (Xu et al., 2020) as the backbone which derives the vector representation of each token $\mathbf{V}^r$ and $\mathbf{V}^c$. The matrix $\Phi_u$ is predicted by dot-multiplying every pair of $\mathbf{V}_i^r$ and $\mathbf{V}_j^c$. LogicMP applies the FOLC to this matrix prediction $\Phi_u$ to constrain the output.

Table 9: The compilation time of AC for transitivity rule w.r.t. sequence length.

| sequence length | #variables | #clauses | compilation time (s) |
|---|---|---|---|
| ... | ... | ... | ... |
| 5 | 25 | 125 | 0.38 |
| 6 | 36 | 216 | 11.1 |
| 7 | 49 | 343 | 135.7 |
| 8 | 64 | 512 | 6419.2 (1.78h) |
| 9 | 81 | 729 | >24h |
| ... | ... | ... | ... |
| 512 | 262K | 134M | ... |

of performing joint inference to update all 262K variables for a sequence length of 512 in just 0.03 seconds.

### J.4 PSEUDO CODE OF LOGICMP FOR TRANSITIVITY RULE

The implementation of LogicMP for the transitivity rule is quite simple which only consists three different tensor operations. The pseudo PyTorch-like code is shown in Algorithm 2. We can use a few lines of code to integrate the transitivity into the logits derived from the encoding neural network. The computation overhead is small as it only involves dense tensor operations. It is also worth noting that the FOLCs can be automatically transformed into the update code by parsing the rules, thus allowing a general logical FOL language for integrating LogicMP easily.

---

**Algorithm 2** PyTorch-like Code for LogicMP with Transitivity Rule

```
# logits:  torch.Tensor, size=[batchsize, nentities, nentities, 2]
# niterations:  int, number of iterations

cur_logits = logits.clone()
for i in range(niterations):
    Q = softmax(cur_logits, dim=-1)
    cur_logits = logits.clone()
    # Message Aggregation for Implication ∀a,b,c: C(a,b) ∧ C(b,c) ⟹ C(a,c)
    msg_to_ac = einsum('kab,kbc->kac', Q[...,1], Q[...,1])
    # Message Aggregation for Implication ∀a,b,c: C(a,b) ∧ ¬C(a,c) ⟹ ¬C(b,c)
    msg_to_bc = - einsum('kab,kac->kbc', Q[...,1], Q[...,0])
    # Message Aggregation for Implication ∀a,b,c: C(b,c) ∧ ¬C(a,c) ⟹ ¬C(a,b)
    msg_to_ab = - einsum('kbc,kac->kab', Q[...,1], Q[...,0])
    msg = msg_to_ac + msg_to_bc + msg_to_ac
    cur_logits[..., 1] += msg * weight
# Returns cur_logits
```

---

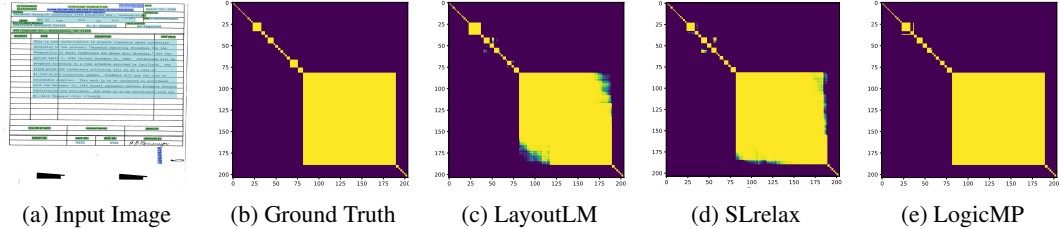

(a) Input Image    (b) Ground Truth    (c) LayoutLM    (d) SLrelax    (e) LogicMP

Figure 8: The visualization of an example that LogicMP can successfully incorporate the FOLC.

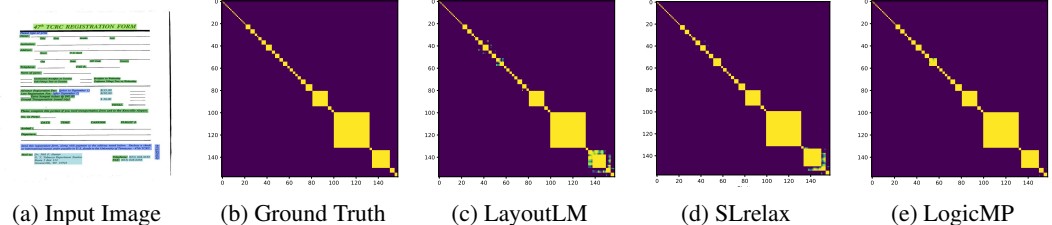

(a) Input Image    (b) Ground Truth    (c) LayoutLM    (d) SLrelax    (e) LogicMP

Figure 9: The visualization of an example that LogicMP can successfully incorporate the FOLC.

### J.5 MORE VISUALIZATION

We illustrate additional three examples in Fig. 8, 9 and 10. In these figures, the output using an independent softmax layer will produce incoherent predictions, i.e., the output is not a series of rectangles. Note that the rectangle structure can be inferred from the predictions of other pairs via the transitivity FOLC. LogicMP can successfully integrate the FOLC to regularize the output as shown in the figures. This FOLC indeed has three meanings: (1) $\forall a, b, c : \mathtt{C}(a,b) \wedge \mathtt{C}(b,c) \implies \mathtt{C}(a,c)$ to enhance the probability of being pair iff $\mathtt{C}(a,b) \wedge \mathtt{C}(b,c)$. (2) $\forall a, b, c : \mathtt{C}(a,b) \wedge \neg\mathtt{C}(a,c) \implies \neg\mathtt{C}(b,c)$ to decrease the probability of being pair iff $\mathtt{C}(a,b) \wedge \neg\mathtt{C}(a,c)$. (3) $\forall a, b, c : \neg\mathtt{C}(a,c) \wedge \mathtt{C}(b,c) \implies \neg\mathtt{C}(a,b)$ to decrease the probability of being pair if $\neg\mathtt{C}(a,c) \wedge \mathtt{C}(b,c)$. LogicMP performs the logical message passing for all three implications and therefore, has the capacity to both (1) remove low-confidence token pairs and (2) fill the missing token pairs.

## K MORE RESULTS ON COLLECTIVE CLASSIFICATION

### K.1 PREDICTION TASKS

There are several predicates in each dataset. In the Kinship dataset, the prediction task is to answer the gender of the person in the query, e.g., $\mathtt{male}(c)$, which can be inferred from the relationship between the persons. For instance, a person can be deduced as a male by the fact that he is the father of someone, and the formula expressing a father is male. In the UW-CSE dataset, we need to infer $\mathtt{AdvisedBy}(a, b)$ when the facts about teachers and students are given. The dataset is split into five sets according to the home department of the entities. The Cora dataset contains the queries to de-duplicate entities and one of the queries is $\mathtt{SameTitle}(a, b)$. The dataset is also split into five

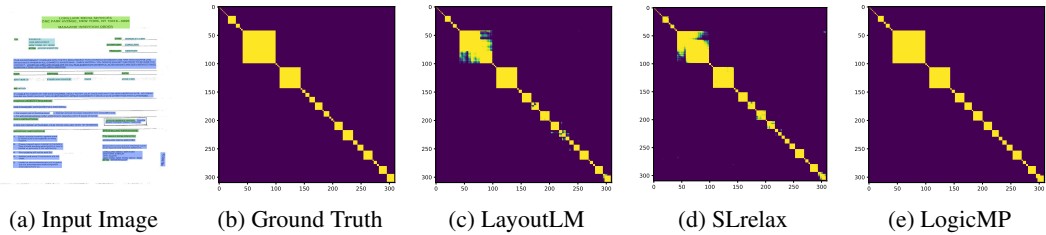

(a) Input Image    (b) Ground Truth    (c) LayoutLM    (d) SLrelax    (e) LogicMP

Figure 10: The visualization of an example that LogicMP can successfully incorporate the FOLC .

subsets according to the field of research. Note this is not the Cora dataset (Sen et al., 2008) typically for the graph node classification.

**Statistics.** The details of the benchmark datasets are illustrated in Table 10.

Table 10: The details of the benchmark datasets.

| Dataset | #entity | #predicate | #observed fact | #query | #ground atom | #ground formula |
|---|---|---|---|---|---|---|
| Kinship/S1 | 62 | 15 | 187 | 38 | 50K | 550K |
| Kinship/S2 | 110 | 15 | 307 | 62 | 158K | 3M |
| Kinship/S3 | 160 | 15 | 482 | 102 | 333K | 9M |
| Kinship/S4 | 221 | 15 | 723 | 150 | 635K | 23M |
| Kinship/S5 | 266 | 15 | 885 | 183 | 920K | 39M |
| UW-CSE/AI | 300 | 22 | 731 | 4K | 95K | 73M |
| UW-CSE/Graphics | 195 | 22 | 449 | 4K | 70K | 64M |
| UW-CSE/Language | 82 | 22 | 182 | 1K | 15K | 9M |
| UW-CSE/Systems | 277 | 22 | 733 | 5K | 95K | 121M |
| UW-CSE/Theory | 174 | 22 | 465 | 2K | 51K | 54M |
| Cora/S1 | 670 | 10 | 11K | 2K | 175K | 621B |
| Cora/S2 | 602 | 10 | 9K | 2K | 156K | 431B |
| Cora/S3 | 607 | 10 | 18K | 3K | 156K | 438B |
| Cora/S4 | 600 | 10 | 12K | 2K | 160K | 435B |
| Cora/S5 | 600 | 10 | 11K | 2K | 140K | 339B |

## K.2 DATASET FORMULAS

We show several logic rules in the datasets in Table 11. The blocks each of which contains 5 rule examples correspond to the Smoke, Kinship, UW-CSE, and Cora datasets. The maximum length of Smoke and Kinship rules is 3, and 6 for the UW-CSE and Cora datasets. We can see from the table that all the logic formulas are CNF. Note that some formulas contain fixed constants such as "Post_Quals" and "Level_100" and we should not treat them as arguments.

Table 11: Several logic rules in the datasets.

| First-order Logic Formula | #Predicates |
|---|---|
| $\neg\texttt{smoke}(a) \vee \neg\texttt{friend}(a,b) \vee \texttt{smoke}(b)$ | 3 |
| $\neg\texttt{smoke}(a) \vee \texttt{cancer}(a)$ | 2 |
| $\texttt{smoke}(a) \vee \neg\texttt{cancer}(a)$ | 2 |
| $\neg\texttt{friend}(a,a)$ | 1 |
| $\neg\texttt{friend}(a,b) \vee \texttt{friend}(a,b)$ | 2 |
| $\neg\texttt{female}(x) \vee \neg\texttt{child}(y,x) \vee \texttt{mother}(x,y)$ | 3 |
| $\neg\texttt{male}(x) \vee \neg\texttt{child}(y,x) \vee \texttt{father}(x,y)$ | 3 |
| $\neg\texttt{female}(x) \vee \neg\texttt{child}(x,y) \vee \texttt{daughter}(x,y)$ | 3 |
| $\neg\texttt{male}(x) \vee \neg\texttt{child}(x,y) \vee \texttt{son}(x,y)$ | 3 |
| $\neg\texttt{male}(x) \vee \neg\texttt{female}(x)$ | 2 |
| $\neg\texttt{taughtBy}(c,p,q) \vee \neg\texttt{courseLevel}(c,\texttt{Level\_500}) \vee \neg\texttt{ta}(c,s,q) \vee \texttt{advisedBy}(s,p) \vee \texttt{tempAdvisedBy}(s,p)$ | 5 |
| $\neg\texttt{publication}(p,x) \vee \neg\texttt{publication}(p,y) \vee \neg\texttt{student}(x) \vee \texttt{student}(y) \vee \texttt{advisedBy}(x,y) \vee \texttt{tempAdvisedBy}(x,y)$ | 5 |
| $\neg\texttt{inPhase}(s,\texttt{Post\_Quals}) \vee \neg\texttt{taughtBy}(c,p,q) \vee \neg\texttt{ta}(c,s,q) \vee \texttt{courseLevel}(c,\texttt{Level\_100}) \vee \texttt{advisedBy}(s,p)$ | 5 |
| $\neg\texttt{student}(x) \vee \texttt{advisedBy}(x,y) \vee \texttt{tempAdvisedBy}(x,y)$ | 3 |
| $\neg\texttt{publication}(t,a) \vee \neg\texttt{publication}(t,b) \vee \texttt{samePerson}(a,b) \vee \texttt{advisedBy}(a,b) \vee \texttt{advisedBy}(b,a)$ | 5 |
| $\neg\texttt{Author}(bc1,a1) \vee \neg\texttt{Author}(bc2,a2) \vee \neg\texttt{HasWordAuthor}(a1,+w) \vee \neg\texttt{HasWordAuthor}(a2,+w) \vee \texttt{SameBib}(bc1,bc2)$ | 5 |
| $\neg\texttt{Author}(bc1,a1) \vee \neg\texttt{Author}(bc2,a2) \vee \texttt{HasWordAuthor}(a1,+w) \vee \neg\texttt{HasWordAuthor}(a2,+w) \vee \texttt{SameBib}(bc1,bc2)$ | 5 |
| $\neg\texttt{Title}(bc1,t1) \vee \neg\texttt{Title}(bc2,t2) \vee \neg\texttt{HasWordTitle}(t1,+w) \vee \neg\texttt{HasWordTitle}(t2,+w) \vee \texttt{SameBib}(bc1,bc2)$ | 5 |
| $\neg\texttt{Title}(bc1,t1) \vee \neg\texttt{Title}(bc2,t2) \vee \texttt{HasWordTitle}(t1,+w) \vee \neg\texttt{HasWordTitle}(t2,+w) \vee \texttt{SameBib}(bc1,bc2)$ | 5 |
| $\neg\texttt{Venue}(bc1,v1) \vee \neg\texttt{Venue}(bc2,v2) \vee \neg\texttt{HasWordVenue}(v1,+w) \vee \neg\texttt{HasWordVenue}(v2,+w) \vee SameBib(bc1,bc2)$ | 5 |

## K.3 GENERAL SETTINGS

The experiments were conducted on a basic machine with a 16GB P100 GPU and an Intel E5-2682 v4 CPU at 2.50GHz with 32GB RAM. The model is trained with the Adam optimizer with a learning rate of 5e-4.

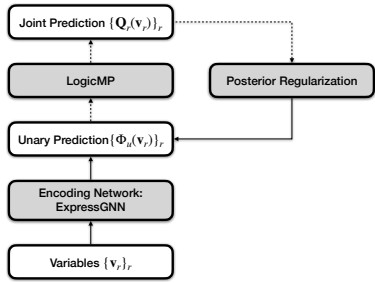

Figure 11: The illustration of incorporating the logic rules into the models via posterior regularization. In this learning paradigm, LogicMP plays the role of logical inference layer for the encoding network. Specifically, we are given a set of variables $\mathbf{v}$. The encoding network takes the variables as input and output scores $\Phi_u$ as the unary potentials of variables. Then the unary potentials are fed into the LogicMP layers to perform MF inference to derive updated marginals $\mathbf{Q}$. They in turn become the targets of the encoding network through the distillation loss. The dotted line means no gradients in the back-propagation.

## K.4 DETAILS OF OUR METHOD

Fig. 11 gives an illustration of our approach. In the figure, LogicMP is stacked upon an encoding network with parameters $\theta$ to take advantage of both worlds (the symbolic ability of LogicMP and the semantic ability of the encoding network). The encoding network is responsible for estimating the unary potentials of the variables independently. Then the LogicMP layers take these unary potentials to perform MF inference which derives a more accurate prediction with the constraints of logic rules. The outputs of LogicMP layers then become the targets of the encoding network through a distillation loss for better estimation. In this way, logical knowledge can be distilled into the encoding network. Note that LogicMP in this process only performs inference without any learning. Intuitively, the encoding network is responsible for point estimation, while LogicMP gives the joint estimation via symbolic reasoning. LogicMP helps to adjust the distribution of the encoding network since the output of LogicMP can not only match the original prediction but also fit the logic rules.

We show the derivation of such a learning paradigm from the posterior regularization (Ganchev et al., 2010) as follows. Specifically, the posterior regularization method derives another target distribution $h(\mathbf{v})$ by minimizing $D_{KL}(h(\mathbf{v})\|p_\theta(\mathbf{v}|O))$ with the prior constraints $\phi$ from the logic rules. The optimal $h(\mathbf{v})$ can be obtained in the closed form: $h(\mathbf{v}) \sim p_\theta(\mathbf{v}|O)\exp(\lambda\phi(\mathbf{v},O))$, where $\lambda$ is a hyper-parameter. When the constraint $\phi(\mathbf{v},O) = \sum_{f\in F} w_f \sum_{g\in G_f} \phi_f(\mathbf{v}_g)$ (Markov logic) and $p_\theta(\mathbf{v}|O) \sim \exp(\sum_i \phi_u(v_i;\theta))$ (encoding network), we have $h(\mathbf{v}) \sim \exp(\sum_i \phi_u(v_i;\theta) + \lambda \sum_{f\in F} w_f \sum_{g\in G_f} \phi_f(\mathbf{v}_g))$ which is equivalent to our definition in Eq. 1. We want to distill $h(\mathbf{v})$ to $p_\theta(\mathbf{v}|O)$. Following the work (Wang et al., 2021), we calculate the marginal of target distribution for each variable $Q_i(v_i)$ via LogicMP and distill the knowledge by minimizing the distance between local marginals and unary predictions, i.e., $\mathcal{L} = \sum_i l(Q_i(v_i), p_\theta(v_i|O))$, where $l$ is the loss function selected according to the specific applications. In the implementation, we use the mean-square error of their logits.

**Proposition K.1.** *The variational E-step training and posterior regularization with LogicMP ($T = 1$) have the same learning objective.*

*Proof.* Simple proof can be obtained by investigating the gradient of each ground atom. In the VIEM method, the posterior probabilistic distributions of ground atoms (denoted as $Q_i(v_i)$) are independent and the learning objective is to maximize the total MLN score of groundings $\sum_f w_f \sum_{g\in G_f} \mathbb{E}_{\mathbf{v}_g} \phi_f(\mathbf{v}_g) = \sum_f w_f \sum_{g\in G_f} \sum_{\mathbf{v}_g} \phi_f(\mathbf{v}_g) \prod_{j\in g} Q_j(v_j)$ (see Eq. 4 in (Zhang et al., 2020)). Take a single grounding as an example, the gradient of $Q_i$ w.r.t. a grounding $g$ is $\frac{\partial \sum_{\mathbf{v}_g} \phi_f(\mathbf{v}_g)Q_i(v_i)\prod_{j\in g_{-i}} Q_j(v_j)}{\partial Q_i(v_i)}$. Combining all groundings together, the gradient becomes $\frac{\partial \sum_f w_f \sum_{g\in G_f(i)} \sum_{\mathbf{v}_g} \phi_f(\mathbf{v}_g)Q_i(v_i)\prod_{j\in g_{-i}} Q_j(v_j)}{\partial Q_i(v_i)} = \sum_f w_f \sum_{g\in G_f(i)} \sum_{\mathbf{v}_g} \phi_f(\mathbf{v}_g)\prod_{j\in g_{-i}} Q_j(v_j)$. It is exactly what each mean-field update computes for each ground atom. With a single iteration, we have the conclusion. $\qquad\square$

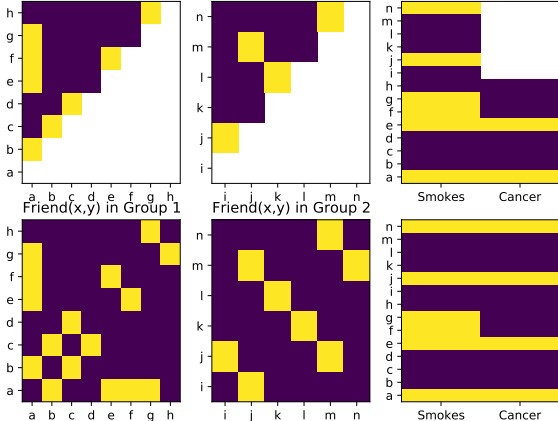

Figure 12: Facts (top) and predictions (bottom) on the Smoke dataset (🟨/■ for the true/false facts).

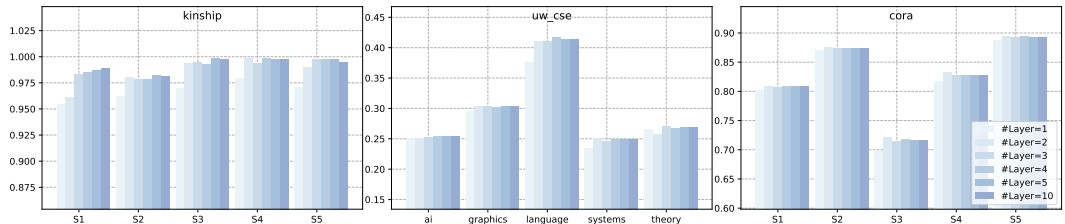

Figure 13: The ablation results on three datasets.

Despite the theoretical equivalence, the implementation with LogicMP in practice is much more scalable due to its parallel computation and thereby permits more training within a reasonable time, leading to significant performance improvement. Besides, LogicMP allows performing multiple iterations to obtain a more precise gradient which also brings performance improvements.

### K.5 MORE EXPERIMENTAL RESULTS

**Smoke dataset.** Smoke is an example dataset for sanity check and Fig. 12 shows that the correct results can be predicted.

**Ablation on the number of iterations.** Fig. 13 represents the ablation results w.r.t. the number of iterations on the Kinship, UW-CSE, and Cora datasets in detail. The results show that the performance improves consistently when using multiple LogicMP layers, and it keeps stable when we further stack the layers. This is reasonable as the MF algorithm typically converges to a stable state within several steps. LogicMP takes a few steps to converge and we can empirically set $T$ to 5. Although more layers lead to more computation, we observed little computational overhead with multiple stacks of LogicMP layers due to the parallel implementation.

**Inference efficiency.** We show the inference time of the methods on UW-CSE in Table 12 under OWA. The methods except for ExpressGNN w/ GS and ExpressGNN w/ LogicMP use CPU due to the inefficiency of GPU implementation. For ExpressGNN w/o LogicMP, the encoding network is a GNN that can be computed in parallel while its learning from MLN is inefficient as the groundings are obtained sequentially and the computation is expensive when consuming groundings. We can see that LogicMP achieves better inference speed than the competitors. Fig. 14 shows the learning curves of LogicMP variations and ExpressGNN w/ GS. LogicMP w/o Opt denotes the LogicMP without the optimization for Einsum. LogicMP w/o OptEinsum+Parallel denotes that the LogicMP is implemented without parallel Einsum and performs the aggregation sequentially. LogicMP w/o OptEinsum+Parallel+RuleOut also omits the technique to remove the expectation calculation in Sec. 3.1. Comparing them with LogicMP, the parallel Einsum brings evident acceleration and other improvements also improve the efficiency. Note that the Einsum optimization is almost free as it can be done in advance within milliseconds for argument size $\leq 6$ (the maximum arity is 6 in the tested

Table 12: Comparison of inference time on UW-CSE. Better results are in bold.

| | Inference Time (minutes) | | | | | |
| --- | --- | --- | --- | --- | --- | --- |
| | A. | G. | L. | S. | T. | avg. |
| MCMC | >24h | >24h | >24h | >24h | >24h | >24h |
| BP | 408 | 352 | 37 | 457 | 190 | 289 |
| Lifted BP | 321 | 270 | 32 | 525 | 243 | 278 |
| MC-SAT | 172 | 147 | 14 | 196 | 86 | 123 |
| HL-MRF | 135 | 132 | 18 | 178 | 72 | 107 |
| ExpressGNN w/ GS (16K) | 14 | 20 | 5 | 7 | 13 | 12 |
| ExpressGNN w/ LogicMP (16K) | **<1** | **<1** | **<1** | **<1** | **<1** | **<1** |
| ExpressGNN w/ GS (20M) | >24h | >24h | >24h | >24h | >24h | >24h |
| ExpressGNN w/ LogicMP (20M) | **101** | **82** | **64** | **85** | **70** | **80** |

datasets). On Cora, the efficiency keeps similar to UW-CSE while the other MLN methods fail to complete within 24 hours.

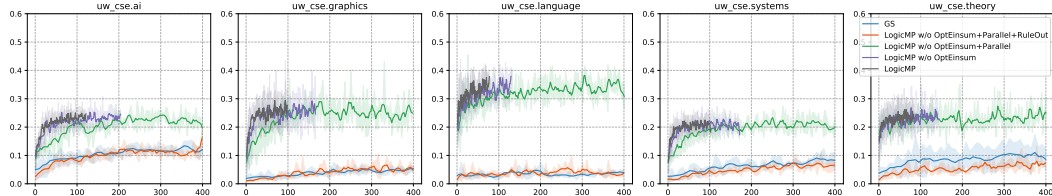

Figure 14: The AUC-PR learning curves w.r.t. minutes of LogicMP variations and VIEM.

