# OpenReview forum: "LogicMP: A Neuro-symbolic Approach for Encoding First-order Logic Constraints"
_ICLR.cc/2024/Conference — ICLR 2024 poster_

### Official Review · Reviewer_9vvz · 2023-10-30

**Soundness:** 2 fair
**Presentation:** 3 good
**Contribution:** 2 fair
**Rating:** 6
**Confidence:** 4

**Summary:**

The paper introduces LogicMP, a neuro-symbolic method designed to efficiently integrate first-order logic constraints (FOLCs) into neural networks. LogicMP performs mean-field variational inference over an MLN, and its computation is paralleled by leveraging the structure and symmetries in MLNs. The authors demonstrate the effectiveness and efficiency of LogicMP through empirical results in various domains. The results show that LogicMP outperforms the baselines in both performance and efficiency.

**Strengths:**

- The paper is well-motivated and easy to follow.
- Using Einsum notation to formalize the message aggregation of first-order logic rules is very interesting.
- The performance of the proposed method is better than previous work.

**Weaknesses:**

- Although LogicMP focuses on encoding FOLs into neural networks, it cannot handle existential quantifiers, which significantly limits its applicability.
- The evaluation appears somewhat limited. Firstly, it only compares 3 neuro-symbolic baselines (SL, its variant SLrelax, SPL). These methods compile the constraint into a probabilistic circuit and then perform exact inference. Given that LogicMP performs approximate inference, comparing it with methods using exact inference seems somewhat unfair. Indeed, some other works encode the constraints and perform efficient approximate inference [1,2]. Lastly, since most previous work encodes propositional logic into neural networks, a more comprehensive evaluation on these previous benchmarks would enhance the paper's comprehensiveness and validity.

[1] DL2: Training and Querying Neural Networks with Logic

[2] Injecting Logical Constraints into Neural Networks via Straight-Through Estimators

**Questions:**

- What is the expressiveness of LogicMP? Can it encode any quantifier-free first-order logic formula? Additionally, the notation in the first paragraph of Section 2 is somewhat confusing. For instance, is a specific structure required for $f$? In other words, should $f$ be represented in the form of a disjunction of logical atoms? Moreover, is the logical atom $\mathtt{C}(e_1, e_2)$ a general form to represent any relation between $e_1$ and $e_2$?
- Why not directly solve problem Eq.1? In problem Eq.1, we can perform the weighted counting in a parallel manner, rather than using sequential generation as required when solving problems Eq.2 and Eq.3. Moreover, quite a few techniques in fuzzy logic can efficiently handle the discrete potential function $\phi_f(\cdot)$, such as translating the disjunction into the product or the minimum.

---

> ### Author Response · Authors · 2023-11-19
> **Authors’ Response (Part I)**
>
> We are grateful for your thorough review and insightful feedback on our paper.
>
> Below is our response, structured to address your concerns more clearly.
>
> # Contribution of Our Paper
>
> In our research, it's crucial to distinguish our innovative methodology from prior work. Previous efforts in the neuro-symbolic field were generally limited to integrating constraints of **propositional logic** -- either dealing directly with propositional logic [1] or converting first-order logic (FOL) into individual propositional groundings [2]. The complexity of **first-order logic (FOL)** reasoning problems escalates significantly in more intricate scenarios. For example, in our first FUNSD example (Sec. 5.1), there are 134M propositional groundings with over 262K ground atoms, resulting in $2^{2.6e5}$ possible worlds. Performing exact inference at such a scale is intractable. This challenge persists even when leveraging weighted model counting techniques, such as advanced arithmetic circuits (ACs) [3, 4], which fail to reduce this to a solvable problem when the number of entities exceeds eight (see App J.3). The underlying difficulty in FOL reasoning involves weighted first-order model counting (WFOMC), a problem proven to be #P-complete for even moderately complicated rules [5]. The treewidth will be exponential for methods that handle WMC, such as SDD [6]. LogicMP sidesteps the need for exact WFOMC by leveraging the variational mean-field algorithm to alleviate the inference difficulty.
>
> Additionally, the difficulty of FOL reasoning does not solely come from the discrete nature of logic rules but also from the sheer volume of groundings. Previous methods may face challenges in dealing with massive groundings, e.g., 600+B groundings in Cora (Sec. 5.2). DL2 [1] adopts fuzzy logic techniques to implement individual propositional logic rules via neural networks, but how to handle such many groundings remains unclear. ST [2] creates a state matrix $f$ for all the groundings and uses it to calculate the grounding satisfaction. However, instantiating all the groundings is computationally expansive and practically infeasible when the problem becomes intricate. When applying ST to our Cora task, the size of $f$ will be around [600B\*$2^L$], where $L$ is the rule length. LogicMP does not require instantiating groundings; all are automatically aggregated in Einsum operations, where the basic computation unit is the grouped ground atoms of each predicate.
>
> Our approach represents a feasible and scalable neuro-symbolic solution that integrates constraints of **first-order logic**. The basic idea to solve this thorny problem is to use the variational mean field method for an approximate solution, which sidesteps the WFOMC problem with several iterative updates. However, merely applying vanilla iterations (which treat propositional groundings individually) remains highly inefficient. To overcome this, we capitalized on the structural symmetries within the groundings of first-order symbolic logic, converting the mean-field iterations of Markov Logic Networks (MLNs) into parallel computations. This breakthrough led to the development of LogicMP, a novel and efficient solution to this long-standing challenge in the field.
>
> 1. Fischer et al. DL2: training and querying neural networks with logic. ICML 2019.
> 2. Yang et al. Injecting logical constraints into neural networks via straight-through estimators. ICML 2022.
> 3. Xu J, Zhang Z, Friedman T, et al. A semantic loss function for deep learning with symbolic knowledge. ICML 2018.
> 4. Ahmed K, Teso S, Chang K W, et al. Semantic probabilistic layers for neuro-symbolic learning. Advances in Neural Information Processing Systems, 2022.
> 5. Nilesh Dalvi and Dan Suciu. The dichotomy of probabilistic inference for unions of conjunctive queries. JACM 2013.
> 6. https://web.cs.ucla.edu/~guyvdb/talks/IJCAI16-tutorial/IJCAI16-tutorial.pdf

---

> > ### Comment · Reviewer_9vvz · 2023-11-22
> > **Reply**
> >
> > Thank you for the response. I appreciate the technical contributions in this paper. However, it's unclear to me why fuzzy logic-based methods, such as DL2 and primal-dual method [1], would fail in these tasks. Despite known limitations, e.g., suffering from shortcuts in grounding, fuzzy logic-based methods typically substitute logical connectives with (sub)differentiable operators like min/max, which don't significantly increase computational costs. Hence, a comparative experiment could further clarify the effectiveness of the proposed method.
> >
> > [1] Nandwani et al. A primal dual formulation for deep learning with constraints. Neurips 2019.

---

> ### Author Response · Authors · 2023-11-19
> **Authors’ Response (Part II)**
>
> # Answers to Specific Questions
>
> ## Q1: Compared Methods in the Evaluation
> 1. Among the propositional logic methods, we compare against SL and SPL in Sec. 5.1, which use ACs to compute WMC in parallel. To the best of our knowledge, they are more scalable than other methods without ACs and are the most scalable propositional methods, even though they fail to complete the task.
> 2. Among the first-order logic methods, we compare against other MLN-based methods in Sec. 5.2, including lifted MLN, HL-MRF, and ExpressGNN. They have demonstrated strong performance and scalability.
>
> Our experiments cover three general tasks over image, graph, and text, demonstrating LogicMP's effectiveness and versatility.
>
> ## Q2: The Incapability of Handling Existential Quantifier
> The inability to use existential quantifiers stems from MLN limitations, which typically consider only universal quantifiers. This issue is common across MLN-based methods. We are exploring attention-based methods to model Skolem forms to address existential quantifiers.
>
> ## Q3: The Expressiveness of LogicMP
> In LogicMP, $f$ can be any FOL in conjunctive normal form (CNF) with universal quantifiers (the disjunction of literals, i.e., clauses, are the items in CNF). $\mathtt{C}(e_1,e_2)$ denotes the "coexist" predicate for entities $e_1$ and $e_2$ in our CV task (Sec. 5.1). Besides this, LogicMP supports predicates with arbitrary numbers of arguments. Table 11 demonstrates several FOL for the task in Sec. 5.2, which also contains unary predicates and ternary predicates, e.g., "taughtBy(c, p, q)" (p is taught by q in the course c).
>
> ## Q4: Eq.1 is Intractable even with WMC
> As "Contribution of Our Paper" mentions, directly solving Eq.1 is theoretically intractable. Specifically, for any FOL that is not liftable, performing FOL reasoning is #P-complete. For such problems, WMC methods, such as SDD, generate a tree with an extremely large treewidth, making computation infeasible. An example is given in the App. J.3, where SDD is used to compile the transitivity rule with various entities. When the number of entities exceeds 8, the compilation becomes invalid.
>
> ## Q5: Using Fuzzy Logic is Insufficient
> Using fuzzy logic mitigates the inference difficulty. However, previous propositional methods that fail to leverage the FOL structural symmetries may face difficulties dealing with massive groundings, e.g., 134M+ groundings in Sec. 5.1 and 600B+ groundings in Sec. 5.2. DL2 [1] proposes a method to implement individual propositional logic rules via neural networks, but how to handle such many groundings remains challenging. LogicMP does not require instantiating the groundings; all are automatically aggregated in Einsum operations that leverage the FOL structural symmetries, where the basic computation unit is the grouped ground atoms of each predicate.

---

> ### Author Response · Authors · 2023-11-22
> **Comparison between fuzzy logic-based methods and EINSUM (Part I)**
>
> Thank you for your comments. We clarify the advantages of using Einsum over fuzzy logic-based methods below.
>
> In previous approaches utilizing fuzzy logic for the grounding of First-Order Logic (FOL), each grounding is treated individually, without the incorporation of Einsum. Typically, these groundings are instantiated propositionally first, followed by the application of fuzzy logic. Although it's feasible to compute the fuzzy logic of these groundings in parallel, the process remains resource-intensive due to the large number of groundings involved.
>
> To illustrate this, we experimented with comparing two methods:
> 1.  loss calculation using instantiated groundings (INSTANTIATION)
> 2.  message calculation using Einsum (EINSUM).
>
> *The key advantage of using Einsum is twofold: it allows for computations on grouped ground atoms represented as dense matrices, and it utilizes dense tensor operations instead of sparse indexing for each grounding*, as demonstrated in the accompanying code snippet (see comment in "# get cxy using sparse indexing").
>
> ## Experiment Setup
> Our tests were conducted on the FUNSD task with the transitivity rule, represented as $\forall x, y, z: \neg c(x, y) \vee \neg c(y, z) \vee c(x, z)$. With $N$ entities, this results in $N^3$ groundings.
>
> The hardware enviroment: GPU: A100 GPU (80GB), CPU: Intel(R) Xeon(R) Platinum 8378A CPU @ 3.00GHz, MEM: 1TB.
>
> ## Results
> As illustrated in Table 1, EINSUM shows remarkable scalability up to 2048 entities (8.6 billion groundings), requiring just three matrix multiplication operations. In contrast, the INSTANTIATION approach, which generates all groundings and calculates loss, becomes exceedingly space-consuming. Beyond 256 entities, this method fails on our A100 GPU with 80GB memory, rendering it impractical for large FOLC problems. Notably, EINSUM outperforms INSTANTIATION significantly: for $N=256$, the time required is 0.0228s versus 0.0012s, making it 19 times faster.
>
> **Table 1: Time Consumption Comparison**
>
> This table shows the time consumption for INSTANTIATION and EINSUM across various entity and grounding counts, highlighting where Out Of Memory (OOM) errors occur for INSTANTIATION at higher entity counts, while EINSUM maintains functionality.
>
> | #ENTITY | #GROUNDINGS | INSTANTIATION | EINSUM   |
> |---------|-------------|---------------|----------|
> | 64      | 262144      | 0.0004s       | 0.0007s  |
> | 128     | 2097152     | 0.0033s       | 0.0008s  |
> | 192     | 7077888     | 0.0095s       | 0.0009s  |
> | 256     | 16777216    | 0.0228s       | 0.0012s  |
> | 320     | 32768000    | OOM           | 0.0009s  |
> | 384     | 56623104    | OOM           | 0.0010s  |
> | 448     | 89915392    | OOM           | 0.0011s  |
> | 512     | 134217728   | OOM           | 0.0012s  |
> | 576     | 191102976   | OOM           | 0.0014s  |
> | 640     | 262144000   | OOM           | 0.0012s  |
> | 704     | 348913664   | OOM           | 0.0012s  |
> | 768     | 452984832   | OOM           | 0.0013s  |
> | 832     | 575930368   | OOM           | 0.0016s  |
> | 896     | 719323136   | OOM           | 0.0016s  |
> | 960     | 884736000   | OOM           | 0.0018s  |
> |1024    | 1073741824  | OOM           | 0.0019s  |
> | ... | ...| ... | ...|
> |2048   | 8589934592 | OOM | 0.0009s|

---

> ### Author Response · Authors · 2023-11-22
> **Comparison between fuzzy logic-based methods and EINSUM (Part II)**
>
> ## Code Implementation
>
> Here, we provide the Python code snippet demonstrating the methods for instantiating groundings, calculating loss using these groundings, and calculating messages using Einsum. It emphasizes the difference in approach between the 'calculate_loss_using_groundings' and 'calculate_message_using_einsum' functions.
>
> ```python
> import torch
> from torch import nn
> import torch.nn.functional as F
> import time
>
> def instantiate_all_groundings(N):
>     xyz = torch.arange(N**3).long().cuda()
>     x = xyz // (N**2)
>     xyz = xyz % (N**2)
>     y = xyz // (N**1)
>     xyz = xyz % (N**1)
>     z = xyz // (N**0)
>     return [x.long(), y.long(), z.long()]
>
> def calculate_loss_using_groundings(logits, groundings):
>     Q = F.softmax(logits, dim=-1)
>     x, y, z = groundings
>     #print(x, y, z)
>
>     # get cxy using sparse indexing
>     cxy = Q[:, x, y, 1]
>     cyz = Q[:, y, z, 1]
>     cxz = Q[:, x, z, 1]
>     #print(cxy)
>
>     # fuzzy logic
>     loss = torch.maximum(torch.maximum((1 - cxy), (1 - cyz)), cxz)
>     return loss
>
> def calculate_message_using_einsum(logits):
>     Q = F.softmax(logits, dim=-1)
>     # c(x, y) and c(y, z) -> c(x, z)
>     _msg1 = torch.einsum('bxy,byz->bxz', Q[...,1], Q[...,1])
>     # c(x, y) and not c(x, z) -> not c(y, z)
>     _msg2 = torch.einsum('bxy,bxz->byz', Q[...,1], Q[...,0])
>     # c(y, z) and not c(x, z) -> not c(x, y)
>     _msg3 = torch.einsum('byz,bxz->bxy', Q[...,1], Q[...,0])
>     msg = _msg1 - _msg2 - _msg3
>     return msg
>
> def main():
>     bs = 128
>     Ns = [32 * i for i in range(1, 1024//32+1)]
>
>     print('By using Einsum')
>     for N in Ns:
>         print(f'Estimating time for {N} entities for {N**3} groundings.')
>         # since computation time in neural network remains the same,
>         # we simply generate one.
>         logits = torch.zeros(bs, N, N, 2).cuda()
>         cur = time.time()
>         calculate_message_using_einsum(logits)
>         print('Time of Calculating: {:.4f}s'.format(time.time() - cur))
>         print()
>
>     print('By instantiating all groundings')
>     for N in Ns:
>         print(f'Estimating time for {N} entities for {N**3} groundings.')
>         logits = torch.zeros(bs, N, N, 2).cuda()
>         cur = time.time()
>         groundings = instantiate_all_groundings(N)
>         print('Time of Instantiating: {:4.4f}s'.format(time.time() - cur))
>
>         cur = time.time()
>         calculate_loss_using_groundings(logits, groundings)
>         print('Time of Calculating: {:4.4f}s'.format(time.time() - cur))
>         print()
>
>
> main()
> ```

---

### Official Review · Reviewer_RzRh · 2023-10-30

**Soundness:** 3 good
**Presentation:** 2 fair
**Contribution:** 1 poor
**Rating:** 5
**Confidence:** 3

**Summary:**

The paper proposes a mean-field variational scheme for inference in Markov Logic Networks. The corresponding message passing scheme exploits some structure of the formulas and a tensor operation to speed-up a naive mean filed approximation.

**Strengths:**

The technique is sound and the paper is generally well-written.
Experiments are diverse.

**Weaknesses:**

The novelty of the paper is limited and cannot be assessed from the current paper. This is a major weakness,

The paper fails in positioning in the wider field of neuro-symbolic AI.

The paper claims to be the first method capable of encoding FOLC (pag. 2, “Contributions”). This is not true. The authors themselves cite ExpressGNN. However, there are many other papers attempting at this. I will cite some here, but many more can be found following the corresponding citations:
Deep Logic Models, Marra et  al, ECML 2019
Relational Neural Machines, Marra et al, ECAI 2020
NeuPSL: Neural Probabilistic Soft Logic, Pryor et al, 2023
DeepPSL: End-to-End Perception and Reasoning, Dasaratha et al, IJCAI 2023
Backpropagating Through MLNs, Betz et al, IJCLR 2021

Many of these systems have CV and citation networks experiments.

**Questions:**

1) The paper mentions FOLC but it never defines them. All the examples, though, are definite clauses. Are non-definite clause supported? If yes, are you employing them in your experiments?

2) Is there an impact in the size of the observed / non-observed split? Usually, there is a great imbalance between the two and it is not clear to me how this may impact the message passing / the pruning of messages.

---

> ### Author Response · Authors · 2023-11-19
> **Authors’ Response**
>
> We sincerely appreciate your invaluable time in reviewing our paper and your comments.
>
> Please find our responses below, and we hope they can address your concerns.
>
> # Contribution of Our Paper
>
> Performing FOL reasoning remains a long-standing problem involving weighted first-order model counting (WFOMC), a problem proven to be #P-complete for even moderately complicated rules [0]. Although many previous works have made efforts in the neuro-symbolic field to combine FOLCs, the problem is not fully addressed. Ideally, we want an approach that (1) can incorporate FOL as constraints, (2) over arbitrary neural networks, and (3) allows end-to-end training via backpropagation. Previous work does not enjoy all of these properties:
>
> - DLM [1] integrates the FOL over a neural network, but the training requires estimating the MAP solution, making it not end-to-end. Property (3) is not satisfied.
> - RNM [2] integrates the FOL over a neural network, but the training of MLN and the neural network remains separate, similar to ExpressGNN. Property (3) is not satisfied.
> - NeuPSL [3] models joint energy for the neural network and symbolic rules and directly optimizes the energy loss. However, in our supervised tasks (sec. 5.1 and 5.3), $y^*$ is always given, and no FOLC will be violated, resulting in a training identical to pure neural training. For other tasks where the $y^*$ is not fully provided, how to obtain $z$ in their Sec.5 remains unclear; it is typically an intractable FOL reasoning problem mentioned above. Property (3) is not satisfied.
> - DeepPSL [4] and learnable MLN [5] represent the first-order logic using neural networks, but the neural backbones for pure neural semantics are not integrated. Property (2) is not satisfied.
> - Other works, such as DeepProbLog [6], Scallop [7], SL [8], and SPL [9], do not employ the FOLC but rather propositional logic. Property (1) is not satisfied.
>
> Our approach effectively addresses this long-standing FOL reasoning problem, marking a feasible and scalable neuro-symbolic solution that integrates first-order logic constraints in arbitrary neural networks via end-to-end backpropagation. We employ the variational mean-field method to address the intractability issue, transforming the problem into several iterative updates. However, merely applying vanilla iterations (which treat propositional groundings sequentially) remains highly inefficient. To overcome this, we capitalized on the structural symmetries within the groundings of first-order symbolic logic, converting the mean-field iterations of Markov Logic Networks (MLNs) into parallel computations. This breakthrough led to the development of LogicMP, a novel and efficient solution to this long-standing challenge in the field.
>
> 0. Nilesh et al. The dichotomy of probabilistic inference for unions of conjunctive queries. JACM 2013.
> 1. Marra et al., Integrating Learning and Reasoning with Deep Logic Models, ECML 2019
> 2. Marra et al., Relational Neural Machines, ECAI 2020
> 3. Pryor et al., NeuPSL: Neural Probabilistic Soft Logic, 2023
> 4. Dasaratha et al., DeepPSL: End-to-End Perception and Reasoning, IJCAI 2023
> 5. Betz et al., Backpropagating Through MLNs, IJCLR 2021
> 6. Manhaeve et al., Deepproblog: Neural probabilistic logic programming. NIPS, 2018.
> 7. Jiani et al., Scallop: From Probabilistic Deductive Databases to Scalable Differentiable Reasoning. NIPS 2021.
> 8. Jingyi et al., A semantic loss function for deep learning with symbolic knowledge. ICML 2018
> 9. Kareem et al., Semantic probabilistic layers for neuro-symbolic learning. NIPS 2022
>
> # Answers to Specific Questions
>
> ## Q1: Claim of "first neuro-symbolic approach capable of encoding FOLCs"
> Please see "Contribution of Our Paper". We acknowledge these important methods in the related work section in the new version. We rephrase the sentence to be more appropriate -- "first fully differentiable neuro-symbolic approach capable of encoding FOLCs for arbitrary neural networks."
>
> ## Q2: Definition of FOLC
> The FOLC was defined in the first line of the second paragraph of the "Introduction" section with an example describing it.
>
> ## Q3: Support of non-definite clause
> LogicMP supports conjunctive normal form (CNF) with universal quantifiers, including definite and non-definite clauses. In the collective classification experiment, we used several non-definite clauses shown in Table 11, such as "¬taughtBy(c, p, q) ∨ ¬courseLevel(c, Level_500) ∨ ¬ta(c, s, q) ∨ advisedBy(s, p) ∨ tempAdvisedBy(s, p)".
>
> ## Q4: The impact in the size of the observed / non-observed split
> In our experiments, the split is provided in the raw tasks. In the collective classification, the #observed / (#non-observed + #observed) is about 0.001/0.01/0.08 in Kinship/UW-CSE/Cora. The observed facts cover only a small portion of the ground atom, making the task difficult. LogicMP outperforms competitors (including HL-MRF and ExpressGNN) by a large margin (+173\% in UW-CSE/+28\% in Cora over ExpressGNN).

---

### Official Review · Reviewer_XeZX · 2023-10-31

**Soundness:** 3 good
**Presentation:** 3 good
**Contribution:** 3 good
**Rating:** 8
**Confidence:** 4

**Summary:**

A scalable inference method is proposed for MLNs using neural networks. The main idea is to use Mean Field iterations to perform approximate inference in MLNs. Further, since this relies on sending messages in a ground MLN which can be very large, messages are aggregated across symmetrical groundings to improve scalability. It is shown that this can be formalized using Einsum summation. The advantage with this approach is that the messages can be computed through parallel tensor operations. Experiments are performed on several different types of problems and comparisons are presented using state-of-the-art methods

**Strengths:**

- The use of Einsum to aggregate and parallelize ground MLN messages in MF seems to be a novel and interesting idea for scaling up inference through neural computations.
- The experiments seem extensive and are performed on a variety of different problems showing generality of the approach

**Weaknesses:**

- In terms of significance, there has been a long history of work in lifted inference with the same underlying principle of using symmetries to scale-up inference in MLNs. One of the key takeaways from such work (e.g. Broeck & Darwiche 2013) is that evidence can destroy symmetries in which case lifted inference reduces to ground inference (if guarantees on the inference results are required). Here, while the approach is scalable, would the same problem be encountered. In the related work section, it is mentioned that for earlier methods, “The latter consists of symmetric lifted algorithms which become inefficient with distinctive evidence”. Does this mean that LogicMP does not have this issue? While the neural approximation can scale-up, I don’t know if there is a principled way to trade-off between quality of inference results (due to approximation using einsum) and scalability. The experiments though show that using LogicMP in different cases yield good results.

**Questions:**

How does evidence affect Einsum computations? Does it break symmetries making it harder to parallelize?

There has been studies in databases regarding width of a FOL (Vardi)  (e.g. in a chain formula, the width is small). This has also been used to scale-up inference using CSPs (Venugopal et al. AAAI 2015, Sarkhel et al. IJCAI 2016). Would this be related to Einsum optimization?

In the experiments were the weights for the MLN formulas encoded by LogicMP learned (it is mentioned in one case that the weights was set to 1). How do these weights impact performance?

---

> ### Author Response · Authors · 2023-11-19
> **Authors’ Response**
>
> We sincerely appreciate your useful comments and your acceptance of our work.
>
> Please find our responses below.
>
> ## Q1: The availability of LogicMP for MLN with distinctive evidence
>
> Yes, we are very excited about this property of LogicMP: it can perform efficient MLN inference **with distinctive evidence**. Note that our formalization in Eq. 1 contains the distinctive evidence, i.e., the first "neural semantics" item ($\sum_i \phi_u(v_i)$). Performing inference over Eq. 1 means performing MLN inference with distinctive evidence. In our first FUNSD experiment, there are 262K ground atoms with distinctive evidence (distinctive neural predictions). Using a modern GPU, LogicMP can perform MLN inference with 262K variables and 134M groundings in just 0.03 seconds.
>
> Let's explain how it is achieved. Since the exact inference for general FOL reasoning (WFOMC) is proved to be #P-complete, our LogicMP achieves efficient inference at the cost of variational approximation. LogicMP breaks down the dependency among the ground atoms by variational inference and uses the mean-field algorithm to iteratively approach the variational approximation of MLN distribution. Even though the mean-field update is theoretically possible, vanilla sequential implementation is impractical -- instantiating and computing 134M groundings is not feasible in practice. The efficiency of LogicMP is essentially achieved by leveraging the structural symmetries in the groundings of FOL and formalizing the message aggregation into Einsum tensor operations, which is principally different from previous lifted methods for WFOMC.
>
>
> ## Q2: Distinct evidence will not affect Einsum computations
>
> LogicMP is designed to cope with the distinctive evidence from the neural networks, so it will **not** affect the computational efficiency of Einsum. LogicMP first computes a marginal distribution $\mathbf{Q}$ for each ground atom using the distinctive evidence (when no evidence, i.e., no neural prediction is provided, zero is used as the evidence). Then, the Einsum operation takes the marginal probability $\mathbf{Q}$ and calculates the grounding messages at the level of FOL in parallel. In this way, the computation remains the same with or without distinctive evidence. The distinctive evidence will destroy the symmetries in the lifted inference but not the structural symmetries among the FOL groundings. Instead of relying on the identity among the groundings, LogicMP leverages the structural symmetries among the FOL groundings, which persists when distinctive evidence is given.
>
> ## Q3: Connection and Comparison with CSP
>
> The relationship between the CSP and Einsum formalization was discussed in the App. H. Concretely, CSP is proposed for grounding counting (GC), while Einsum is for grounding message aggregation (GMA). GC is similar to GMA as both can be formalized as the summation of the product. CSP optimizes GC by reducing the problem into sub-problems, and it is related to Einsum optimization -- both Einsum optimization and CSP are equivalent to the classical ML problem of variable elimination in the message passing.
>
> Even though CSP has several differences from our approach:
>
> 1. Unlike GC, the Einsum handles GMA derived from the mean-field algorithm.
> 2. More importantly, we propose Einsum as a **parallel tensor operation** that brings computational efficiency and enables LogicMP to be an effective neural module, which is not emphasized in CSP. Using Einsum, MLN inference can be expressed as an efficient parallel NN.
>
> ## Q4: Learning and the Effect of Rule Weights
>
> The weights are learned via back-propagation in the supervised tasks (CV FUNSD and NLP sequence labeling tasks). We found that the weights were not sensitive to the initialization during the experiments. The learned weight converges to ~0.1 for various initializations in the FUNSD task. This is reasonable as the balance between the FOLC and neural network is automatically achieved during the training. For the collective classification task in Sec. 5.2, we adopt a fashion of semi-supervised learning where the rule is used to enhance the neural networks and set the weight to 1. In this setup, the amount of weight is a coefficient of the learning rate in the training. In our experiments, we found it robust.

---

> > ### Comment · Reviewer_XeZX · 2023-11-23
> >
> > Thanks for your response.

---

### Official Review · Reviewer_svJm · 2023-11-01

**Soundness:** 2 fair
**Presentation:** 1 poor
**Contribution:** 2 fair
**Rating:** 5
**Confidence:** 2

**Summary:**

The paper proposes a novel neural layer, called LogicMP, which can be plugged into any off-the-shelf neural network to encode constraints expressed in First Order Logic.

**Strengths:**

Relevance:

The paper deals with a very important problem that is of interest to the larger AI community.

Novelty:

The paper introduces a novel layer. However, it fails to acknowledge other works that have integrated logical constraints into a neural network layer. Among the most relevant we find:
- Nicholas Hoernle, Rafael-Michael Karampatsis, Vaishak Belle, and Kobi Gal. MultiplexNet: Towards fully satisfied logical constraints in neural networks. In Proc. of AAAI, 2022.
- Eleonora Giunchiglia and Thomas Lukasiewicz. Multi-label classification neural networks with hard logical constraints. JAIR, 72, 2021.
- Tao Li and Vivek Srikumar. Augmenting neural networks with first-order logic. In Proc. of ACL, 2019.

**Weaknesses:**

Clarity:

Overall, I found the paper not very readable, and I think the authors should try to give more intuitions.
See below for some questions I had while reading the paper.

- While the authors included an overview of Markic Logic Networks there are still some concepts that look a bit obscure. What does the weight associated with each formula represent? Is it a way of representing the importance assigned to the formula? Why do the authors need the open-world assumption? When explaining the MLNs, can you please add an example with: (i) an ML problem, (ii) what would the FOLC be in the problem, and (iii) what would be the observed and unobserved variables in the problem.

- What does it mean that $Z_i$ is the partition function? Over what?

- I am not sure how to read Table 1. Same applies for Figure 3.

- How is it possible that the complexity does not depend on the number of formulas $|F|$?

- Finally, are the constraints guaranteed to be satisfied or they are just incorporated?

**Questions:**

See above.

---

> ### Author Response · Authors · 2023-11-19
> **Authors’ Response (Part I)**
>
> We sincerely thank you for your helpful suggestions and comments.
>
> Please find our responses below.
>
> # Contribution of Our Paper
>
> In light of our research, it's essential to distinguish our innovative methodology from prior work. Previous efforts in the neuro-symbolic field were primarily limited to integrating constraints of **propositional logic** — either dealing with propositional logic [1,2] or converting first-order logic (FOL) into individual propositional groundings [3]. However, considering more intricate scenarios, the complexity of **first-order logic** reasoning escalates significantly. For instance, our first FUNSD example has over 262K ground atoms. Performing inference among $2^{2.6e5}$ possible worlds is intractable. This challenge persists even when leveraging the advanced arithmetic circuits (ACs) [4, 5], which fail to reduce this to a solvable problem when the number of entities exceeds eight. This underlying difficulty of FOL reasoning involves weighted first-order model counting (WFOMC), a problem proven to be #P-complete for even moderately complicated rules [6].
>
> Our approach is an efficient and scalable neuro-symbolic solution that integrates constraints of **first-order logic** over arbitrary neural networks. The basic idea to solve this thorny problem is to use the variational mean field method for an approximate solution, which sidesteps the WFOMC problem with several iterative updates.
> Even though, merely applying vanilla iterations (which treat propositional groundings individually) remains highly inefficient. To overcome this, we capitalized on the structural symmetries within the groundings of first-order symbolic logic, converting the mean-field iterations of Markov Logic Networks (MLNs) into parallel computations. This led to the development of LogicMP, a novel and efficient solution to this long-standing challenge of FOL reasoning.
>
> 1. Nicholas Hoernle, Rafael-Michael Karampatsis, Vaishak Belle, and Kobi Gal. MultiplexNet: Towards fully satisfied logical constraints in neural networks. In Proc. of AAAI, 2022.
> 2. Eleonora Giunchiglia and Thomas Lukasiewicz. Multi-label classification neural networks with hard logical constraints. JAIR, 72, 2021.
> 3. Tao Li and Vivek Srikumar. Augmenting neural networks with first-order logic. In Proc. of ACL, 2019.
> 4. Xu J, Zhang Z, Friedman T, et al. A semantic loss function for deep learning with symbolic knowledge. ICML 2018.
> 5. Ahmed K, Teso S, Chang K W, et al. Semantic probabilistic layers for neuro-symbolic learning. Advances in Neural Information Processing Systems, 2022.
> 6. Nilesh Dalvi and Dan Suciu. The dichotomy of probabilistic inference for unions of conjunctive queries. JACM 2013.
>
> # Answers to Specific Questions
>
> ## Q1: Acknowledgement of Related Work
>
> We have added the references of these important methods in the related work section of the new version.
>
> ## Q2: Markov Logic Networks (MLNs)
>
> Markov Logic Networks merge classical Markov networks with a potential function quantifying how well a state adheres to given rules. The key is in weighting these rules: each rule is assigned a weight that signifies its importance. The higher the weight, the more significant its impact on the potential function, suggesting greater importance. In extreme cases, a high weight effectively turns a rule into a hard constraint.
>
> ## Q3: Open-World Assumption Necessity
>
> In the general tasks where logic is integrated, making assumptions about the states of unobserved ground atoms is problematic. Since each atom could be either True or False, relying on the closed-world assumption is impractical. For example, in our first case study (Sec. 5.1), all instances of $\mathtt{C}(e_i,e_j)$ are unobserved. However, each instance has a non-negligible probability of being True, rendering the closed-world assumption unsuitable.

---

> ### Author Response · Authors · 2023-11-19
> **Authors’ Response (Part II)**
>
> ## Q4: MLN Example
>
> Task Overview: Consider an image document containing both image pixels and text tokens (e.g., an image with the text "Date: February 25, 1998 ... CLIENT TO: L8557.002 ..."). The objective here is to segment these tokens into blocks (such as grouping "February 25, 1998" into a single block).
>
>    - ML Problem Formalization: We are presented with a dataset $\mathbf{D}$ comprising various samples. Each sample includes a tuple $(\mathbf{e}, \mathbf{m}, \mathbf{V})$, where $\mathbf{m}$ denotes the input image, $\mathbf{e} = \{e_1, ..., e_N\}$ represents $N$ input token IDs, and $\mathbf{V} = \{\mathtt{C}(e_i, e_j)\}_{i, j}$ is an $N\times N$ matrix indicating whether tokens $e_i$ and $e_j$ coexist in the same block. The goal is to develop a function $f: (\mathbf{e}, \mathbf{m}) \rightarrow \mathbf{V}$ that maps this input to a label space $\mathbf{V}$. In our experiments, this function is implemented by LayoutLM+LogicMP and trained on $\mathbf{D}$ to optimize the likelihood.
>    - Observations: These include the input elements, namely the image $\mathbf{m}$ and the token IDs $\mathbf{e}$.
>    - Latent Variables: These are the binary variables we aim to predict, represented as $\mathbf{V}=[v_{\mathtt{C}(1, 1)}, ...., v_{\mathtt{C}(1, N)}, ..., v_{\mathtt{C}(N, 1)}, ..., v_{\mathtt{C}(N, N)}]$.
>    - FOLC of 'Transitivity': If tokens $i$ and $j$ are in the same block and tokens $j$ and $k$ are also together, then tokens $i$ and $k$ should be in the same block. This is formally stated as "$\forall i, j, k; \mathtt{C}(i, j) \wedge \mathtt{C}(j, k) \rightarrow \mathtt{C}(i, k)$" where $\mathtt{C}$ denotes the $coexist$ predicate.
>
>
> ## Q5: The Partition Function $Z_i$
>
> The partition function $Z_i$ is the summation of potentials for two states of $v_i$: $v_i=0$ and $v_i=1$. Specifically, it's calculated as
> $Z_i = \exp(\phi_u(v_i=0) + \sum_{f\in F} w_f \sum_{g\in G_f(i)} \hat{Q}\_{i, g}(v_i=0)) + \exp(\phi_u(v_i=1) + \sum_{f\in F} w_f \sum_{g\in G_f(i)} \hat{Q}_{i, g}(v_i=1))$.
> This approach, in contrast to Eq.1 sums over $2^{|V|}$ items, simplifies the computation by only considering two scenarios, greatly easing the computational burden.
>
> ## Q6: Interpreting Table 1 and Figure 3
>
> Table 1 demonstrates the influence of the grounding $\neg \mathtt{C}(e_1,e_2) \vee \neg \mathtt{C}(e_2,e_3) \vee \mathtt{C}(e_1,e_3)$ on $\mathtt{C}(e_2, e_3)$. It corresponds to the implication: $\mathtt{C}(e_1,e_2) \wedge \mathtt{C}(e_2,e_3) \implies \mathtt{C}(e_1,e_3)$. The table shows that when the premise $\mathtt{C}(e_1,e_2) \wedge \mathtt{C}(e_2,e_3)$ is False, i.e., $(\mathtt{C}(e_1,e_2), \mathtt{C}(e_2,e_3)) = (0, 0), (0, 1)$ or $(1, 0)$, the rule holds regardless of the hypothesis ($\mathtt{C}(e_2,e_3)$) being True or False, implying that in such states, the grounding does not influence the hypothesis, allowing us to skip certain computations.
>
> Figure 3 visualizes the parallel message passing for the transitivity rule. It comprises three parallel operations based on different implication statements:
>    1. The first row in the update (grey area) corresponds to $\forall a, b, c: \mathtt{C}(a, b) \wedge \mathtt{C}(b,c) \implies \mathtt{C}(a, c)$, calculated using $\mathtt{einsum}(\text{``}ab,bc\to ac\text{''}, {\mathbf{Q}}\_{\mathtt{C}}(\mathbf{1}), {\mathbf{Q}}_{\mathtt{C}}(\mathbf{1}))$.
>    2. The second row corresponds to $\forall a, b, c: \mathtt{C}(a, b) \wedge \neg \mathtt{C}(a,c) \implies \neg \mathtt{C}(a, c)$, calculated using $\mathtt{einsum}(\text{``}ab,ac\to bc\text{''}, {\mathbf{Q}}\_{\mathtt{C}}(\mathbf{1}),{\mathbf{Q}}_{\mathtt{C}}(\mathbf{0}))$.
>    3. The third row corresponds to $\forall a, b, c: \mathtt{C}(b, c) \wedge \neg \mathtt{C}(a,c) \implies \neg \mathtt{C}(a, b)$, calculated using $\mathtt{einsum}(\text{``}bc,ac\to ac\text{''}, {\mathbf{Q}}\_{\mathtt{C}}(\mathbf{1}),{\mathbf{Q}}_{\mathtt{C}}(\mathbf{0}))$.
>
> The red frame highlights the relation between a grounding message's calculation in the sequential process and the parallel computation.
>
>
> ## Q7: Complexity Notation
>
> The most intricate rule typically dominates complexity in our LogicMP. Therefore, we represent overall complexity w.r.t. the most complex rule, which provides a practical measure of the system's computational demands. The complexity improvement remains the same when including $|F|$.
>
>
> ## Q8: Constraint Satisfaction in MLNs
>
> In Markov Logic Networks, rules are not inherently guaranteed to be satisfied; they are designed to balance various rules and observed evidence. A rule is more likely to be satisfied if it has a high weight. In extreme cases, a high weight effectively turns a rule into a hard constraint.

---

### Meta-Review · Area_Chair_be6S · 2023-12-09

**Metareview:**

This work tackles the problem of scaling inference in neuro-symbolic (nesy) systems comprising a neural network backbone plust some symbolic first order logic represented as rules. To this end, the authors propose to use a mean-field approximation for inference in Markov Logic Networks (MLNs) embedded as last layers of a neural encoder, and called LogicMP. The proposed LogicMP is evaluated on a battery of nesy tasks on relational data.

The reviewers overall appreciated the contribution and the scaling of the proposed approximate reasoning scheme. At the same time they argue that presentation was lacking context on the background, missing experimental details and overclaiming the novelty of the contribution.
The authors incorporated much of this feedback in an updated revision and improved the presentation, toning down claims, and adding missing details. I believe the contribution can be valuable for the nesy community as a good baseline for FO constraints, even if it boils down to a (cleverly) engineered inference pipeline.

**Justification For Why Not Higher Score:**

The contribution is at its core engineering a fast approximation that relies on previous theory.

**Justification For Why Not Lower Score:**

I believe the contribution is still meaningful in the rich nesy literature. If downstream task performance is the key in a nesy task, then a fast and simple approximation that is still accurate w.r.t. the downstream task is valuable.

---

### Decision · Program_Chairs · 2024-01-16

Accept (poster)